# Structure of the human heparan sulfate polymerase complex EXT1-EXT2

**Francisco Leisico** [1,5], **Juneina Omeiri** [1,5], **Christine Le Narvor**[2], **Joël Beaudouin**[1], **Michael Hons** [3], **Daphna Fenel**[1], **Guy Schoehn** [1], **Yohann Couté** [4], **David Bonnaffé** [2], **Rabia Sadir**[1], **Hugues Lortat-Jacob** [1] ✉ & **Rebekka Wild** [1] ✉

Heparan sulfates are complex polysaccharides that mediate the interaction with a broad range of protein ligands at the cell surface. A key step in heparan sulfate biosynthesis is catalyzed by the bi-functional glycosyltransferases EXT1 and EXT2, which generate the glycan backbone consisting of repeating *N*-acetylglucosamine and glucuronic acid units. The molecular mechanism of heparan sulfate chain polymerization remains, however, unknown. Here, we present the cryo-electron microscopy structure of human EXT1-EXT2, which reveals the formation of a tightly packed hetero-dimeric complex harboring four glycosyltransferase domains. A combination of in vitro and *in cellulo* mutational studies is used to dissect the functional role of the four catalytic sites. While EXT1 can catalyze both glycosyltransferase reactions, our results indicate that EXT2 might only have *N*-acetylglucosamine transferase activity. Our findings provide mechanistic insight into heparan sulfate chain elongation as a nonprocessive process and lay the foundation for future studies on EXT1-EXT2 function in health and disease.

Heparan sulfates (HS) are long and complex polysaccharide chains found on the surface of almost all animal cells[1,2]. They are covalently attached to the serine residue of a core protein, together forming the so-called proteoglycan[3–5]. Proteoglycans can interact with a broad range of cellular factors, including chemokines, growth factors, cytokines, morphogens, signaling receptors, adhesion molecules, pathogens and others, thereby influencing their local concentration, stability and conformation[6,7]. This tremendous functional versatility is associated with the structural complexity of HS, whose composition was shown to depend on the cell type and developmental stage[8]. Malfunctioning of HS biosynthesis, on the other hand, has been linked to an increased susceptibility to Alzheimer's disease, inflammation, viral infections, and tumorigenesis[9–12].

Biosynthesis of HS takes place in the endoplasmic reticulum and Golgi lumen and starts with the assembly of a tetra-saccharide linker (glucuronic acid-galactose-galactose-xylose)[13]. After the transfer of a *N*-acetylglucosamine (GlcNAc) moiety, the long HS polysaccharide backbone is polymerized by the alternating addition of glucuronic acid (GlcA) and GlcNAc molecules. Subsequent modifications of the polysaccharide backbone by GlcNAc *N*-deacetylation and *N*-sulfation, GlcA C5 epimerization and 2-, 3- and 6- *O*-sulfation complete the HS biosynthesis process[5,6]. Apart from EXT1, EXT2 and the *N*-deacetylase/*N*-sulfotransferases NDST1-4, which have resisted structural analysis up to now, most enzymes involved in HS biosynthesis have been structurally and functionally characterized[13–17].

EXT1 and EXT2, from exostosin-1 and −2, were originally described as tumor suppressors following the observation that mutations in the corresponding genes were found in patients suffering from hereditary multiple exostoses, a disorder characterized by cartilaginous and bone tumors[18–20]. They were next shown to carry out HS chain elongation and early studies indicated that both enzymes are bi-functional, harboring GlcA- and GlcNAc-transferase activities[21–23]. EXT1 and EXT2 are

[1]Institut de Biologie Structurale, UMR 5075, University Grenoble Alpes, CNRS, CEA, 38000 Grenoble, France. [2]Université Paris-Saclay, CNRS, Institut de chimie moléculaire et des matériaux d'Orsay, 91405 Orsay, France. [3]European Molecular Biology Laboratory (EMBL), Grenoble Outstation, 71 avenue des Martyrs, 38042 Grenoble, France. [4]University Grenoble Alpes, INSERM, CEA, UMR BioSanté U1292, CNRS, CEA, FR2048, 38000 Grenoble, France. [5]These authors contributed equally: Francisco Leisico, Juneina Omeiri. ✉e-mail: Hugues.Lortat-Jacob@ibs.fr; Rebekka.Wild@ibs.fr

predicted to share a common architecture with a N-terminal transmembrane helix anchoring the enzymes in the Golgi membrane, followed by the glucuronic acid transferase (GlcA-T) domain and the C-terminal *N*-acetylglucosamine transferase (GlcNAc-T) domain. Despite a potential redundancy in catalytic activity, both enzymes are required for proper HS polymerization and it was suggested that EXT1 and EXT2 form a functional unit[24]. In such a complex, EXT2, which only features a low catalytic activity by itself, was proposed to increase the activity of EXT1 or to assist in proper folding and transport of EXT1 from the ER to the Golgi[24,25].

The molecular basis for HS biosynthesis, the architecture of a potential hetero-dimeric EXT1-EXT2 complex and the contribution of the two bi-functional glycosyltransferases to the chain polymerization reaction remained unknown. Here we report a high-resolution cryo-electron microscopy (cryo-EM) structure of the human EXT1-EXT2 complex in combination with in vitro and *in cellulo* functional analysis, providing a molecular insight into EXT1-EXT2 complex assembly, the mechanism of donor substrate recognition, and suggesting a non-processive mode for HS chain elongation.

## Results

### In vitro activity of purified human EXT1-EXT2

EXT1 and EXT2, lacking the N-terminal transmembrane helix, were co-expressed as a secreted complex using human embryonic kidney cells (FreeStyle 293-F). We purified 0.5 mg of the complex from 900 mL suspension culture using two immobilized metal affinity chromatography steps and an interspersed protease treatment to cleave off the alkaline phosphatase fusion protein serving as a secretion signal. Although previous studies suggested that EXT1 and EXT2 can be expressed on their own[21,24], we were only able to purify the complex upon co-expression of both proteins. Expression of EXT1 or EXT2 alone resulted in almost undetectable amounts of secreted protein, supporting the view, that the two proteins need to associate in order to be correctly folded and/or transported through the secretory pathway[24,25]. SDS-PAGE and mass spectrometry-based proteomic analysis confirmed the quality of the sample with EXT1 and EXT2 found to be the most abundant and present in comparable amounts (Fig. 1a, Supplementary Table 1). Next, we studied the oligomeric state of the proteins using mass photometry. The main peak was observed at 160 kDa, suggesting that the majority of purified protein particles form dimers in solution (Fig. 1b). To follow the bi-functional activity of the EXT1-EXT2 complex in vitro, we developed a fluorophore-assisted carbohydrate electrophoresis (FACE) assay[26] using chemically and chemo-enzymatically synthesized fluorescent oligosaccharide acceptor substrate analogs (Fig. 1c, Supplementary Fig. 1 and 2). The octa-saccharide dp8 (GlcA-GlcNAc)$_4$ and nona-saccharide dp9 GlcNAc-(GlcA-GlcNAc)$_4$ substrates, carrying a Alexa Fluor 430 at the reducing end, were readily accepted by EXT1-EXT2 as acceptor substrates. The complex catalyzed the specific addition of a single GlcNAc or GlcA moiety onto the dp8 and dp9 acceptor substrates, respectively. While the transfer of a GlcNAc residue onto dp8 led to a slower migration rate due to the increase in size, the addition of a GlcA moiety onto dp9 enhanced the migration rate. This phenomenon can be explained by the higher negative charge of the dp10 reaction product. Conversely, incubation of the EXT1-EXT2 complex with dp9 and UDP-GlcNAc or with dp8 and UDP-GlcA did not result in substrate elongation, emphasizing the high specificity of the two glycosyltransferase reactions. Consistently, the generation of longer polysaccharide chains was only observed in the presence of both UDP-GlcNAc and UDP-GlcA donor substrates (Fig. 1c).

### Architecture of EXT1-EXT2

Single-particle cryo-EM studies on EXT1-EXT2, in presence of both UDP-sugar donor substrates and divalent manganese ions, provided a three-dimensional EM reconstruction with an overall resolution of

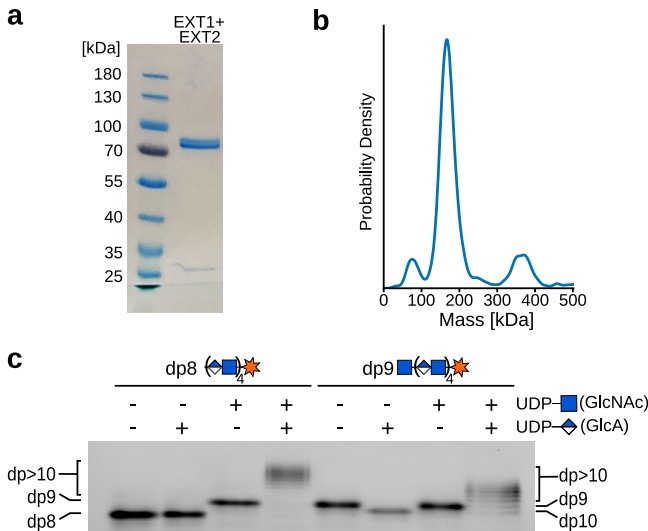

**Fig. 1 | Purification and functional characterization of the EXT1-EXT2 complex. a** Coomassie-stained SDS-PAGE analysis of purified EXT1-EXT2 complex, lacking the N-terminal membrane anchoring helix. Expected molecular sizes of EXT1 and EXT2 are about 85 kDa and 79 kDa, respectively. Source data is provided as a Source Data 1. **b** Mass photometry analysis of purified complex with a major peak at 160 kDa. Source data is provided as a Source Data 2. **c** The catalytic activity of EXT1-EXT2 was studied using an in vitro glycosylation assay. The specific transfer of *N*-acetylglucosamine (GlcNAc) and glucuronic acid (GlcA) molecules was followed using synthetic, fluorescently-labeled octa-saccharide (dp8) and nona-saccharide (dp9) substrate analogs. Source data is provided as a Source Data 3.

2.8 Å (Fig. 2a, Supplementary Fig. 3 and 4, Supplementary Table 2). We de novo built an almost complete model of EXT1-EXT2, only lacking the presumably flexible N-terminal stem regions (residues 29-88 in EXT1, residues 47-77 in EXT2) connecting the catalytic Golgi-luminal domain to the transmembrane anchoring helices, as well as several short, surface-exposed loops. EXT1 and EXT2 form a tightly packed 1:1 hetero-dimeric complex with a total interaction surface of 3523 Å$^2$, suggesting that EXT1-EXT2 complex formation is essential for protein stability. Many charged residues contribute to the assembly of the complex and we found the EXT1 interaction surface to have an overall positive charge, while the one from EXT2 harbors more negatively charged residues (Supplementary Fig. 5a, b). This observation is in line with the theoretical isoelectric points of pH 9.1 and pH 5.7 for the Golgi-luminal parts of EXT1 and EXT2, respectively. Of note, there were no monomeric or homo-dimeric protein particles observed during cryo-EM analysis (Supplementary Fig. 3b, c). The complex spans an approximate dimension of 100 Å x 80 Å x 90 Å, thereby reaching far into the Golgi lumen, which has an average thickness of 100–200 Å[27]. The N-terminal glycosyltransferase domains of EXT1 (residues 89-435) and EXT2 (residues 78-419), supposedly harboring GlcA-T activity, adopt a characteristic GT-B fold architecture[28]. The C-terminal glycosyltransferase domains, formed by residues 479-728 in EXT1 and residues 455-702 in EXT2, feature a GT-A fold and potentially harbour GlcNAc-T activity (Fig. 2b, c, Supplementary Fig. 5c–f). EXT1 and EXT2 share 34% sequence identity and their GlcA-T and GlcNAc-T domains superimpose well with an observed R.M.S.D of 1.2 Å for 209 out of 291 residues and 0.96 Å for 150 out of 182 residues, respectively (Supplementary Fig. 5e, f). Interestingly, the GlcA-T domains of EXT1 and EXT2 interact with each other in a pseudo-C$_2$ symmetric manner, the same observation is made for the GlcNAc-T domains. Differences in the hinge-region between the GlcA-T and GlcNAc-T domains of the two proteins disturb a C$_2$ symmetry axis in the overall complex structure. The GlcA-T and GlcNAc-T domain of EXT1 interact tightly with each other, while the two domains of EXT2 are only connected through a potentially flexible hinge region. The interaction surface between the

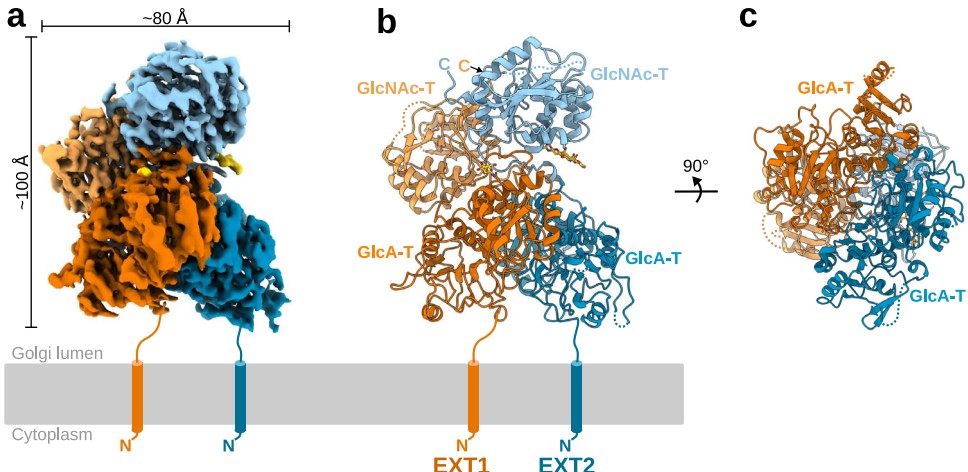

**Fig. 2 | Structure of the human EXT1-EXT2 complex. a** EM map of the hetero-dimeric EXT1-EXT2 complex, with density covering EXT1 in orange and EXT2 in blue. Both proteins harbor a glucuronic acid transferase (GlcA-T) domain and a *N*-acetylglucosamine transferase (GlcNAc-T) domain colored in darker and lighter shades, respectively. Densities of *N*-glycans are colored in yellow. The N-terminal membrane anchoring helices, absent in our expression constructs, are drawn as cylinders. **b** Cartoon representation of the complex with coloring as in **a** and gly-cosylations shown in stick representation. Missing loops are indicated as dotted lines. **c** Rotated view showing the GlcA-T domains from the membrane plane.

GlcA-T domains of EXT1 and EXT2 is similar in size to the one between the GlcNAc-T domains, suggesting that both glycosyltransferase domains contribute equally to hetero-dimeric complex formation. Comparison of our experimental structure with AlphaFold2 models of the individual EXT1 and EXT2 proteins shows that the domain folds are highly similar. Differences include the re-arrangement of the EXT2 GlcNAc-T and GlcA-T domains in respect to each other and the pre-sence of an anti-parallel beta-sheet formed between the C-termini of EXT1 and EXT2. These structural features are correctly predicted by AlphaFold2 when generating a model for the hetero-dimeric complex (Supplementary Fig. 6). Two *N*-linked glycans were found in the EM map, one at the surface of EXT1 (Asn330) and the other in the hinge region of EXT2 (Asn637)(Fig. 2a, b, Supplementary Fig. 5c, d).

## Substrate recognition in the four active sites

EXT1 and EXT2 contain two glycosyltransferase domains each, which presumably harbor GlcA-T or GlcNAc-T activity. To investigate the role of the four catalytic domains in more detail, we studied the topology and the localization of conserved amino acid residues in the active sites. When analyzing the GlcA-T active site of EXT1, we found an additional EM density that fits a UDP ligand (Fig. 3a, Supplementary Fig. 7a, b). The UDP is bound by several charged residues (Arg280, Arg346, Glu349), through a pi-stacking interaction with Tyr324 and a hydrogen bond with Tyr319. Although UDP-GlcA was added during sample preparation, we did not observe any density for the GlcA moiety, which might have been cleaved-off by spontaneous hydrolysis[29]. Two large densities for the uracil ring and the pyropho-sphate group were observed, which allowed to determine the orien-tation of the UDP, the electron density around the ribose ring had poorer quality. Examination of GT-B fold glycosyltransferase struc-tures shows that UDP furanose rings adopt either the $^2E$ or the $^3T_4$ conformation. These two conformations differ mainly in the position of the C3 and O3 atoms, which could explain the lower electron density in this part of the EM map (Supplementary Fig. 7a, b). Another expla-nation could be that the UDP ligand is either bound in sub-stoichiometric amounts or that it can slightly move in the binding pocket. The crystal structure of the protein *O*-glucosyltransferase 1 (POGLUT1, PDB-ID: 5L0U) in complex with a non-hydrolyzable UDP-glucose (UDP-Glc) analog[29] showed that the backbone atoms of loop Gly273 to Phe278 play an important role in the specific recognition of the Glc moiety. Based on structural similarities between the donor substrate binding pockets of POGLUT1 and EXT1, we suggest that the

loop Gly339 to Phe345 in EXT1 is involved in recognizing the GlcA moiety (Supplementary Fig. 8). Previous studies on POGLUT1 pro-posed the involvement of Asp133 in acceptor substrate activation[29]. Interestingly, we found two aspartate residues in EXT1 (Asp162 and Asp164) which could play a similar role (Fig. 3a, Supplementary Fig. 8e, f). Surprisingly, the active site of the EXT2 GlcA-T domain displayed a different topology, with an anti-parallel beta-sheet intruding into the donor substrate binding pocket (Fig. 3b, Supplementary Fig. 7c, d). Furthermore, the arginine residue Arg346, found to be involved in UDP binding in EXT1, as well as two nearby aspartate residues (Asp162 and Asp164 in EXT1) are not conserved in EXT2. Consistently, we did not observe any extra density for a UDP ligand in the active site of EXT2. These structural differences suggest that the GlcA-T domain of EXT2 might not be able to carry out glucuronic acid transfer, hereafter referred to as pseudo-GlcA-T domain.

Next, we studied the active sites of the EXT1 and EXT2 GlcNAc-T domains. Although the UDP-GlcNAc donor substrate was added during cryo-EM sample preparation, we did not observe any well-ordered density for it (Fig. 3c, e). To learn more about the molecular basis of substrate recognition, we performed a search for the closest structural homolog using the DALI server[30]. The highest similarity was observed for the structures of EXTL2 (PDB-ID: 1ON6) and EXTL3 (PDB-ID: 7AUA), two *N*-acetylglucosamine glycosyltransferases involved in the initial steps of HS biosynthesis (Supplementary Fig. 9 and 10)[14,31–33]. EXTL3 has a similar multi-domain composition as the EXT1-EXT2 complex. In contrast to the non-symmetrical EXT1-EXT2 complex, the EXTL3 homo-dimer adopts a $C_2$ symmetry, resulting in a distinct overall architecture (Supplementary Fig. 10). The GlcNAc-T domains of EXT1, EXT2, and EXTL3 show a high degree of structural similarity. The most apparent structural differences between EXT1 and EXTL3 are found in the GlcA-T domain close to the UDP binding site. Of note, EXTL3 has no GlcA transferase activity[33]. Next, we compared EXT1 and EXT2 with the UDP-GlcNAc bound EXTL2 structure. Their superposition reveals a strong spatial conservation of all important residues involved in UDP-GlcNAc donor substrate and metal ion binding, including the DxD motif formed by Asp565 and Asp567 in EXT1 and Asp538 and Asp540 in EXT2, respectively. The GlcNAc moiety is likely to form an extensive hydrogen bond network with several charged residues (Glu653, Asp654, and Arg701 in EXT1; Glu627, Asp628 and Arg673 in EXT2) and a hydrophobic interaction with Leu620 in EXT1, and Leu594 in EXT2, together ensuring the specific recognition of the sugar donor (Fig. 3c–e)[14,34]. Based on these structural similarities, EXT1 and EXT2

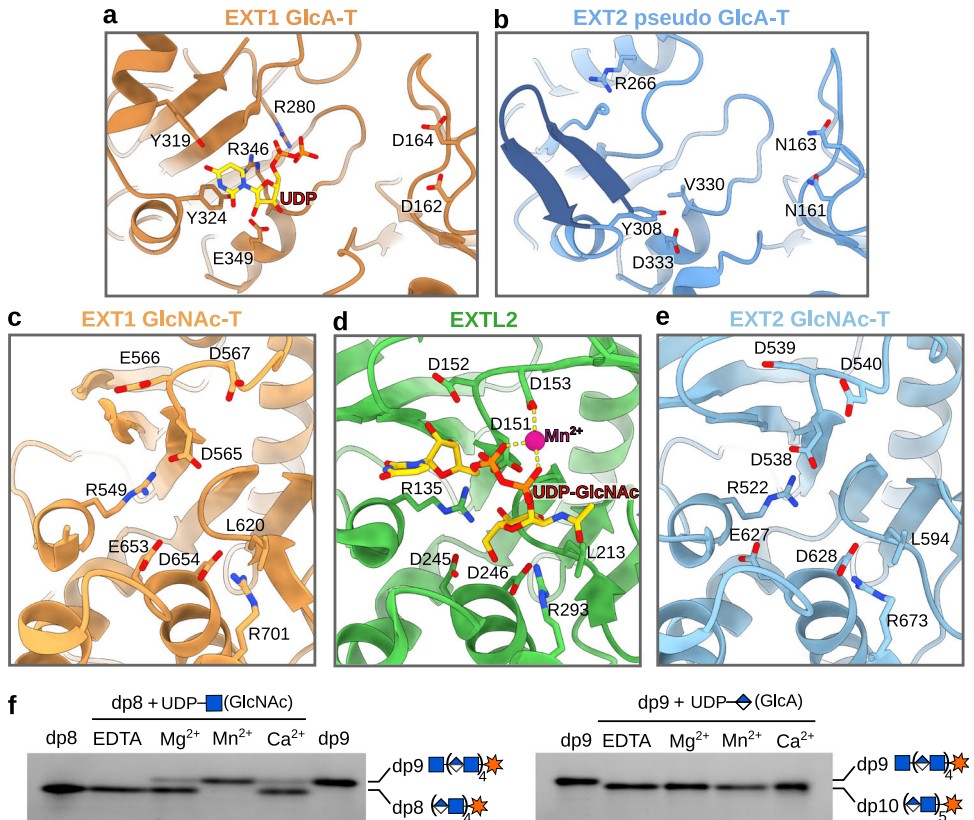

**Fig. 3 | The active sites of the EXT1-EXT2 complex. a** Cartoon representation of the EXT1 GlcA-T catalytic site shown in dark orange with important residues shown as sticks. **b** Catalytic site of the EXT2 pseudo-GlcA-T domain colored in dark blue and shown from the same view as in (a). Two anti-parallel beta-strands, colored in dark blue, intrude into the active site. **c–e** Cartoon representation of the GlcNAc-T active sites of EXT1 (light orange), EXTL2 (green), and EXT2 (light blue) with a UDP-GlcNAc donor substrate (yellow) in the crystal structure of EXTL2 (PDB-ID: 1ON6)[14]. Conserved residues involved in metal ion (shown as purple sphere) or UDP-GlcNAc binding are highlighted as sticks. **f** In vitro activity of EXT1-EXT2 in presence of different divalent metal ions. Fluorescently-labeled octa- (dp8) and nona- (dp9) oligosaccharide substrate analogs were used as acceptor substrates. Source data is provided as a Source Data 4. GlcNAc-T, *N*-Acetylglucosamine-transferase; GlcA-T, glucuronic acid-transferase.

can be presumed to catalyze GlcNAc transfer, with retention of the configuration at the anomeric center, through a $S_N$i-like or orthogonal reaction mechanism, as previously reported for EXTL2[14,35].

To further characterize the importance of metal ions for the GlcA and GlcNAc transfer reactions, we performed in vitro glycosyltransferase assays. In agreement with our structural information, we found the GlcA-T reaction to be metal ion independent, while GlcNAc-T activity required the presence of a divalent metal ion, preferably manganese (Fig. 3f).

**Mutational analysis**

To characterize the catalytic activity of the four glycosyltransferase domains independently and to validate our structural model, we performed in vitro and *in cellulo* mutational analysis. We designed EXT1-EXT2 mutant complexes harboring amino acid substitutions either in the GlcA-T domain (EXT1 D162N/D164N, EXT1 R280A, EXT1 R346A, EXT2 R266A) or the GlcNAc-T domain (EXT1 D565N/D567N and EXT2 D538N/D540N)(Fig. 4a). To ensure that mutations did not affect complex stability, we measured thermal unfolding of purified wild-type and mutant EXT1-EXT2 complexes using nano differential scanning fluorimetry. Melting temperatures ranged between 51 to 55 °C with the EXT1 D565N/D567N containing complex showing a slightly reduced thermal stability (Supplementary Fig. 11). Negative stain EM analysis on wild-type and EXT1 D162N/D164N containing complexes also confirmed sample integrity (Supplementary Fig. 11d, e). In vitro activity assays using FACE demonstrated that the EXT1 D162N/D164N, EXT1 R280A and R346A mutants had a drastically reduced ability to

transfer glucuronic acid, while the EXT2 R266A mutant retained GlcA-T activity (Fig. 4b). This observation confirms that the N-terminal GT-B fold domain harbors GlcA-T activity and further suggests that, under the used reaction conditions, wild-type EXT2 has a low catalytic activity itself. This is in line with the observation that the two aspartate residues important for EXT1 GlcA-T activity are not conserved in EXT2, which displays asparagine residues (Asn161 and Asn163) at these positions (Fig. 3a, b). A decrease in GlcNAc glycosyl transferase activity in the EXT1 D565N/D567N and EXT2 D538N/D540N mutants confirms that the C-terminal GT-A domains of EXT1 and EXT2 harbor GlcNAc-T activity and emphasize the important role of the DxD motif (Fig. 4b).

Next, we set out to study the functional role of the different glycosyltransferase domains of EXT1 and EXT2 in the cellular context. We generated CRISPR-Cas9 HeLa cell lines lacking either the EXT1 or EXT2 protein and quantified the cell surface HS content using an anti-HS antibody and flow cytometry. HS biosynthesis was almost fully abolished in both knock-out cell lines (Fig. 4c, Supplementary Fig. 12), in line with previous studies[36]. To dissect the effect of point mutations in EXT1 or EXT2 on HS biosynthesis, we developed a complementation-based strategy. Generated EXT1 and EXT2 knock-out cell lines were transfected with plasmids encoding either the full-length wild-type protein or mutant versions thereof, and FLAG-tagged EXT1 or EXT2 and HS levels were quantified 36 h post-transfection. Complementation with wild-type EXT1 or EXT2 restored HS levels to around 70 and 45 %, respectively, compared to wild-type HeLa cells. The same observation was made for the EXT2 R266A and EXT2 N637A mutant (Fig. 4c, Supplementary Fig. 12), supporting the view that the pseudo-

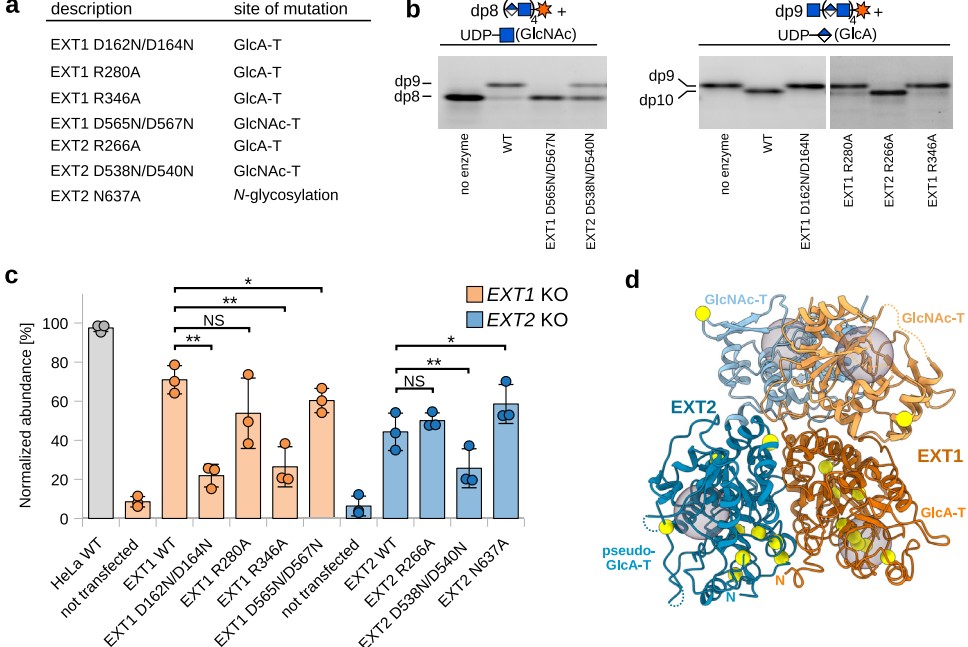

**Fig. 4 | In vitro and in cellulo mutational analysis. a** Table summarizing EXT1 and EXT2 mutants used in this study. **b** In vitro activity assay using fluorescently-labeled octa- (dp8) and nona- (dp9) oligosaccharide acceptor substrates and purified EXT1-EXT2 complexes harboring either EXT1 and EXT2 wild-type proteins (WT) or amino acid substitutions in one of the two proteins. Source data is provided as a Source Data 5. **c** Quantification of cell surface heparan sulfate levels by flow cytometry experiments in HeLa knock-out (KO) cell lines lacking either the EXT1 or EXT2 protein. The knock-out cell lines were not transfected or complemented with either WT EXT1 or EXT2 proteins or amino acid substituted versions thereof. Quantified heparan sulfate content from three independent experiments was normalized to the EXT expression level (see Supplementary Fig. 10) and blotted as percentage compared to wild-type HeLa cells ($n = 3$), shown as dots and average as bars. Error bars indicate SD. A one-sided paired student $t$-test was used to determine $p$-values: NS > 0.05, *<0.05 and **<0.01 or more precisely EXT1 WT vs D162N/D164N (0.0058), R280A (0.0559), R346A (0.0020) and D565N/D567N (0.0485) and EXT2 WT vs R266A (0.1313), D538N/D540N (0.0089) and N637A (0.0178). Source data is provided as a Source Data 6. **d** Mapping of EXT1 and EXT2 missense mutations in patients with hereditary multiple exostoses. EXT1-EXT2 structure is shown as cartoon representation and substituted residues are highlighted as yellow spheres. The four active sites are indicated as gray spheres. GlcNAc-T, *N*-Acetylglucosamine-transferase; GlcA-T, glucuronic acid-transferase.

GlcA-T domain of EXT2 does not participate to HS chain elongation and indicating that the glycan at position 637 does not alter the overall complex characteristics. In contrast, the EXT1 GlcA-T mutants R280A, although not statistically significant, showed reduced HS levels. The EXT1 D162N/D164N and EXT1 R346A mutants had the most striking effect, experiencing almost abolished HS biosynthesis, further demonstrating that the GlcA-T activity of the EXT1-EXT2 complex exclusively relies on EXT1. Alterations in the GlcNAc-T active site of either EXT1 (EXT1 D565N/D567N) or EXT2 (EXT2 D538N/D540N) only partially reduced cell surface HS content, suggesting that both catalytic sites can catalyze GlcNAc addition independently of each other and that both are essential to reach normal complex activity. While the *in cellulo* experiments showed a lower amount of HS for the EXT2 D538N/D540N mutant, we observed a stronger effect for the EXT1 D565N/D567N mutant in vitro. This difference could be explained by the slightly reduced thermal stability of the purified EXT1 D565N/D567N protein complex.

To summarize, a combination of in vitro and *in cellulo* mutational analysis confirmed our structural model and revealed the functional importance of each of the glycosyltransferase domains during HS biosynthesis. Interestingly, we found that the EXT1 GlcA-T domain is essential for complex activity, while mutations in the EXT2 pseudo-GlcA-T domain do not affect HS chain elongation (Fig. 4b, c). Together with our structural analysis suggesting that the EXT2 pseudo-GlcA-T active site cannot accommodate the UDP-activated donor substrate (Fig. 3b, Supplementary Fig. 7c, d), we propose that only EXT1 has GlcA-T activity. GlcNAc transfer, in contrast, seems to be assured by both enzymes.

Mutations in either the *EXT1* or *EXT2* gene in humans are associated with hereditary multiple exostoses, a skeletal disorder characterized by formation of osteochondromas[37]. We compiled a list of all known single amino acid patient mutations linked to a loss-of-function of EXT1 and EXT2 and mapped them onto the complex structure. Interestingly, almost all mutations in EXT1 locate to the GlcA-T active site (Fig. 4d, Supplementary Table 3), again emphasizing the essential role of EXT1 in GlcA transfer and the inability of EXT2 to compensate for the reduced catalytic activity. Patient mutations in EXT2 are mostly located in the pseudo-GlcA-T domain, but not in its catalytic site. These mutations are likely affecting the folding and integrity of the hetero-dimeric complex (Fig. 4d).

## Discussion

EXT1 and EXT2 are two homologous Golgi-localized enzymes, both harboring a GlcA-T and a GlcNAc-T domain. Our cryo-EM structure of EXT1·EXT2 reveals that the two proteins tightly interact to form a stable hetero-dimeric complex. Regions in the cryo-EM map that were poorly resolved are located at the surface of the complex in close proximity to the substrate binding pockets (Supplementary Figs. 3 and 9). High flexibility of loops that are involved in substrate binding is a common feature of many glycosyltransferase structures determined by X-ray crystallography and cryo-EM, especially in the absence of their substrates[14,38,39]. This flexibility ensures easy access of the donor and acceptor substrates to the active site, as well as the release of the reaction products[28,40].

Using structure-based point mutants, we were able to assign the GlcA-T and GlcNAc-T activities to the corresponding catalytic sites and

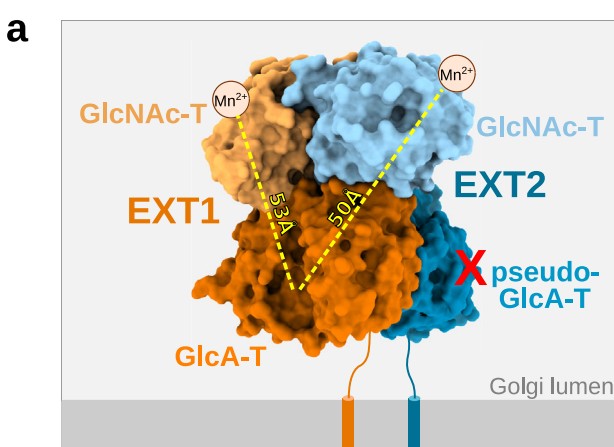

**a**

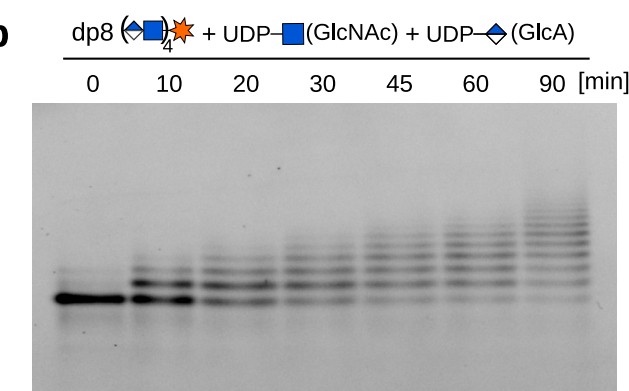

**b**

dp8 + UDP—■(GlcNAc) + UDP—◆(GlcA)

0    10    20    30    45    60    90  [min]

**Fig. 5 | Heparan sulfate chain elongation is nonprocessive. a** Surface representation of the EXT1-EXT2 complex. The metal-ion dependence of the GlcNAc-T reaction is indicated by small spheres with a $Mn^{2+}$ label. A red cross illustrates that the pseudo-GlcA-T domain of EXT2 is catalytic inactive. Distances between the EXT1 GlcA-T active site and the GlcNAc-T active sites of EXT1 and EXT2 are marked by a yellow dotted line. **b** Time course experiment using an excess of fluorescently-labeled oligosaccharide (dp8) substrate and limiting amounts of purified EXT1-EXT2 complex. A ladder-like pattern suggests that GlcNAc and GlcA addition by the EXT1-EXT2 complex occur in a step-wise fashion. Source data is provided as a Source Data 7. GlcNAc-T, N-Acetylglucosamine-transferase; GlcA-T, glucuronic acid-transferase.

demonstrate the molecular determinants of HS biosynthesis. Interestingly, our structural and functional analysis suggest that the EXT2 pseudo-GlcA-T domain is catalytic inactive and that glucuronic acid transfer might be carried out solely by EXT1 (Fig. 4b, c). Our results also provide a reasoning for the observation that known mutations in hereditary multiple exostoses patients, which revealed an accumulation of mutations in the catalytic site of the EXT1 GlcA-T domain (Fig. 4d).

The high structural similarities of the GlcNAc-T active sites of EXT1 and EXT2 with EXTL2, support a $S_N$i-like reaction mechanism for GlcNAc transfer. Currently, there are no close structural homologs of the GlcA-T domain of EXT1 available from which mechanistic insights could be readily derived. $S_N$2-like inverting GTs are characterized by the presence of a base catalyst, which deprotonates and thus activates the C4-hydroxyl group of the acceptor substrate for its nucleophilic attack on the C1 carbon atom of the donor substrate[41]. Structural comparison of EXT1 GlcA-T and POGLUT1 suggests that either Asp162 or Asp164 might play this role, although smaller structural rearrangements in the active site would need to take place (Supplementary Fig. 8). The arginine residues Arg280 and Arg346 are in close proximity

to the pyrophosphate moiety of the UDP and are likely to stabilize the negative charge of the leaving group. Together, with the spatial constraints of the donor and acceptor binding pocket, our observations could suggest a $S_N$2-like mechanism.

Interestingly, the EXT1-EXT2 architecture suggests that HS chain elongation is a nonprocessive reaction as its active sites are located on opposite sites of the complex and, in addition, are facing away of each other (Fig. 5a). Such a spatial arrangement, with a distance of 50 Å between the active sites, is in line with a distributive, multi-step process, during which the elongated polysaccharide chain detaches from EXT1-EXT2 before entering into the next catalytic site. A time course experiment following the chain elongation in vitro further supports this hypothesis as we observed the formation of oligosaccharides of various lengths, rather than the direct conversion of the oligosaccharide substrate to a fully extended HS chain, as it would be expected in case of a processive chain elongation (Fig. 5b)[42].

Assembly of Golgi-localized glycosyltransferases into homo- or hetero-oligomeric complexes is an ancient mechanism, which was proposed to increase protein stability and enzymatic activity[43,44]. Further, an environment-dependent transition between homo- and hetero-oligomers has been described[25]. We present multiple lines of evidence that EXT1 and EXT2 exclusively exist in the form of a hetero-dimeric complex with a 1:1 stoichiometry. First, we were only able to purify secreted EXT1 and EXT2 from the cell medium, when co-expressing both proteins. Mass photometry experiments suggested that the majority of proteins form a dimeric complex in solution (Fig. 1b). In addition, we did not observe any monomeric or homo-dimeric species of EXT1 or EXT2 during cryo-EM analysis (Supplementary Fig. 3).

This raises questions about the functional importance of EXT1-EXT2 complex formation as well as on the organization of the catalytic sites. Combining GlcNAc and GlcA transferase activity into a single enzyme is shortening the substrate traveling path, thereby increasing the efficiency of HS biosynthesis that consists of repeating cycles of GlcA and GlcNAc addition. The presence of two active sites that catalyze GlcNAc transfer, in close proximity to each other, likely further increases the efficiency of chain elongation. Given the importance of proper HS biosynthesis, the generation of a hetero-dimeric EXT1-EXT2 complex might have represented an evolutionary advantage that reduces the deleterious effect of mutations in one of these genes. In this context, the absence of a functional GlcA-T catalytic site in EXT2 is surprising. Sequence analysis of EXT2 proteins of different species, ranging from sea anemone to catfish, drosophila and humans, with 49% sequence identity between the first and the latter, and thus covering 500 million years of evolution, did not reveal any EXT2 sequences, in which the asparagine residues Asn161 and Asn163 were still aspartates, as in the homologous EXT1. This suggests, that the hetero-dimeric EXT1-EXT2 complex and the distribution of its GlcNAc-T and GlcA-T catalytic sites were already present in the last common ancestor of the eumetazoan lineage, thus originating very early on in evolution, possibly from a homo-dimeric EXT complex.

Our human EXT1-EXT2 cryo-EM structure provides molecular insight into HS chain polymerization. Structural and mutational analysis revealed the importance of residues in the active sites and the mode of donor substrate recognition. We were further able to propose a $S_N$i-like mechanism for GlcNAc-T reaction and we presume that GlcA-transfer could follow a $S_N$2-like reaction mechanism. To obtain a complete understanding of the GlcA-T and GlcNAc-T reaction mechanisms, high-resolution structures of EXT1-EXT2 bound to their two acceptor substrates will be needed. Moreover, recombinant EXT1-EXT2 can be exploited to generate oligosaccharides of defined length, such as the nona-saccharide described in this work, a useful tool to study the specific interaction between proteins and HS. The gained mechanistic insight will facilitate in vivo studies on the functional and pathological roles of HS. Finally, this work paves the way for the future

development of structure-based drugs to modulate HS biosynthesis with potential application in antiviral or antitumor treatments[45,46].

## Methods

### Expression and purification of human EXT1-EXT2
For soluble, secreted expression, the coding regions of N-terminal truncated EXT1 (aa29-aa746) and EXT2 (aa47-aa718) were cloned into the pTT22-SSP4 vector (National Research Center Canada, Biotechnology Research Institute) using NheI and BamHI restriction sites. The construct contained a N-terminal secreted alkaline phosphatase fusion protein, followed by a tobacco etch virus (TEV) cleavage site, EXT1 or EXT2, and a C-terminal 8x His tag[47]. 900 mL of Freestyle 293-F cells (Invitrogen, catalog number R79007) were co-transfected with both vectors using polyethyleneimine and incubated at 37 °C for 96 h. All subsequent steps were carried out either at 4 °C or on ice. The supernatant was recovered by centrifugation, sterile filtered, and 50 mM Tris pH 7.8, 150 mM NaCl, 10 mM imidazole, and 10 mM MgCl$_2$ were added, before loading onto a HisTrap HP column (GE Healthcare). After washing with 5CV of buffer A (50 mM Tris pH 7.8, 150 mM NaCl) supplemented with 50 mM imidazole, protein was eluted with buffer A containing 500 mM imidazole. 2 mg of Strep-tagged TEV protease were added to the eluate and the sample was dialyzed for 16 h in buffer A. Cleaved-off alkaline phosphatase and TEV protease were removed by re-purifying EXT1-EXT2 using a second HisTrap HP purification step and by passing protein sample through 1 mL Strep-Tactin XT resin (IBA Lifesciences). Buffer was exchanged by several rounds of concentration and dilution with buffer B (20 mM HEPES pH 7.2, 150 mM NaCl) and the final protein concentration was 0.5 – 0.8 mg mL$^{-1}$. Aliquots were snap-frozen in liquid nitrogen and stored at −80 °C. EXT1 and EXT2 constructs carrying point mutations were generated using site-directed mutagenesis PCR (Supplementary Table 4) and purified as the wild-type complex.

### Chemical synthesis of octa-saccharide substrate analog
The synthetic Alexa Fluor 430™-coupled octa-saccharide (synth dp8-Alexa Fluor 430) was prepared in seven steps from a protected dp8 (Supplementary Fig. 1). The later was prepared by oligomerization of a unique dp2 precursor protected by orthogonal allyl and p-methoxybenzyl groups at the reducing and non-reducing ends, respectively. A dp2 donor was first prepared by removal of the allyl group followed by activation as trichloroacetimidate, while a dp2 acceptor was obtained by acidic cleavage of the p-methoxybenzyl group. Trimethylsilyl triflate promoted coupling of dp2 donor and acceptor in diethylether/tetrahydrofurane (9/1) and gave the expected dp4, bearing also allyl and p-methoxybenzyl groups at the reducing and non reducing ends respectively[48,49]. Repeating, on the dp4, the above-described oligomerization procedure, gave the targeted protected dp8 amenable for further deprotection and functionnalization. After functionalization of the allyl moiety of the fully protected dp8 (GlcA-GlcN$_3$)$_4$ using a thiolene coupling reaction[50] followed by oxidation of the thioether linkage to sulfone with oxone, the azido groups of the dp8 were reduced using 1,3-propanedithiol, and then N-acetylated using anhydride acetic in pyridine at room temperature. Saponification of the methyl esters was performed using aqueous KOH in the presence of H$_2$O$_2$ in a nBuOH/THF mixture. Hydrogenolysis using Pd(OH)$_2$ (20% on charcoal) in a phosphate buffer/t BuOH mixture yielded the amino dp8 compound as a sodium salt after filtration, desalting on a G-25 column, ion exchange on Na+ resin and lyophilization. Then synthetic dp8-Alexa Fluor 430 was obtained by coupling an aqueous solution of amino dp8 with commercial Alexa Fluor 430 NHS ester solubilized in DMSO in the presence of NEt$_3$ overnight at room temperature. Purification using a Sephadex G25 column, eluting with H$_2$O, gave the desired compound. The different oligosaccharide intermediates were characterized by $^1$H, $^{13}$C NMR spectroscopy. Analytical (HPLC), $^1$H, and $^{13}$C NMR data of synthetic dp8-Alexa Fluor 430 are given in Supplementary Fig. 2.

### Generation of nona-saccharide substrate analog
The synthetic Alexa Fluor 430™-coupled octa-saccharide was extended by one GlcNAc moiety using purified EXT1-EXT2. A 200 μL reaction mixture containing 40 μM octa-saccharide, 10 mM UDP-GlcNAc, 0.625 μM EXT1-EXT2, and 10 mM MnCl$_2$ was prepared in buffer B and incubated 16 h at 30 °C. Completion of the reaction was confirmed by fluorophore-assisted carbohydrate electrophoresis (FACE) and the reaction product was purified by size exclusion chromatography using two Superdex Peptide 10/300 GL columns (Cytiva) mounted in a row and pre-equilibrated in 25 mM Tris pH 7.5, 247 mM NaCl and 3 mM KCl. Peak fractions were pooled and dialyzed three times against Mili-Q dH$_2$O using slide-a-lyzer dialysis cassettes with a 2 kDa cut-off (Thermo Fisher Scientific). The desalted sample was lyophilized and resuspended in 100 μl Mili-Q dH$_2$O, resulting in a final concentration of 33.3 μM, as determined by spectrophotometry using a theoretical extinction coefficient of ε = 15,000 cm$^{-1}$ M$^{-1}$ at 430 nm.

### In vitro glycosyl transfer assays
Glycosyltransferase activity of purified EXT1-EXT2 was analyzed by FACE. To study the specificity of GlcA or GlcNAc transfer, 10 μL reaction mix were prepared containing either 5 μM fluorescently-labeled octa- or nona-saccharide acceptor substrate, 0.5 μM purified EXT1-EXT2, 1 mM UDP-GlcA (Sigma) and/or 1 mM UDP-GlcNAc (Sigma) and 10 mM MnCl$_2$ in buffer B. After 60 min incubation at 30 °C, the reaction was stopped by heating the sample for 5 min at 70 °C. 2 μL of glycerol were added to sample prior to loading onto a 25% tris-glycine polyacrylamide gel. Bands were detected using a GelDoc Imaging System (Biorad). The effects of metal ions on GlcA and GlcNAc transfer were analyzed using similar reaction conditions, but either 10 mM EDTA, 10 mM MgCl$_2$, 10 mM MnCl$_2$ or 10 mM CaCl$_2$ were added. GlcNAc transfer was followed by using a reaction mixture containing octasaccharide acceptor substrate and UDP-GlcNAc, GlcA transfer using nona-saccharide acceptor substrate and UDP-GlcA. For mutational analysis, similar reaction conditions were used, but the reaction time was reduced to 30 min at 30 °C and an enzyme concentration of 0.125 μM was used. For the time course experiment, 0.125 μM purified EXT1-EXT2 was mixed with 5 μM fluorescently-labeled octasaccharide, 1 mM UDP-GlcNAc, 1 mM UDP-GlcA and 10 mM MnCl$_2$ in buffer B. Samples were taken at different time points and analyzed by gel electrophoresis.

### Cryo-EM sample preparation and data collection
0.4 mg mL$^{-1}$ purified EXT1-EXT2 was mixed with 1 mM of UDP-GlcNAc, 1 mM UDP-GlcA and 1 mM MnCl$_2$ and incubated for 15 min on ice before applying 4 μL of the sample onto a glow discharged Quantifoil holey carbon grid (R1.2/1.3, 300 mesh, copper). Excess liquid was removed by blotting for 5 s and grid was plunge-frozen in liquid ethane cooled by liquid nitrogen using a Vitrobot Mark IV (FEI), with the chamber temperature set to 4 °C and 100% humidity. Grids were screened on a 200 kV Glacios (FEI) electron microscope. Cryo-EM data was collected at the ESRF facility using beamline CM01[51] on a 300 kV Titan Krios (FEI) electron microscope equipped with a Quantum-LS filter and a K3 direct detection camera (Gatan) using the software EPU (Thermo Fischer Scientific). 7046 movies with 45 frames were acquired in super-resolution mode with a total electron dose of around 46 e$^-$ Å$^{-2}$, at a nominal magnification of 105,000x, corresponding to a pixel size of 0.42 Å at the specimen level. All micrographs were collected with a defocus ranging between −3.0 and −1.0 μm (Supplementary Table 2).

### Image processing
Data processing was carried out in Relion3.1.3[52,53]. The collected movies were corrected for beam-induced motion and binned by a factor of 2 using the MotionCor2 (MotionCor2_1.4.0_Cuda110) frame alignment software[54]. CTFFIND v.4.1 was used to estimate the contrast transfer function (CTF) of non dose-weighted micrographs[55]. After

manual sorting, 5828 micrographs were retained for further processing (Supplementary Fig. 3). A total number of 2'189'505 particles were picked using template-free auto-picking, and fourfold binned particles were extracted and subjected to 2D classification. Best 2D classes showing protein complex from different orientations were used as templates for reference-based auto-picking, resulting in 2,703,248 particles. After several rounds of 2D classification a total number of 516,360 particles were retained. Different strategies to generate an ab initio three-dimensional model resulted in poorly defined maps. To circumvent difficulties in aligning particles, we generated a low-resolution map using a predicted structural model of EXT1-EXT2. For this, a hetero-dimeric complex structure was predicted using the software AlphaFold2[56,57] and converted into a low resolution EM map using the EMAN2[58] command *pdb2mrc* and a resolution of 12 Å. 3D refinement using the generated low-resolution map as a reference, resulted into a 3D reconstruction at a nominal resolution of 3.2 Å. Different 3D classification strategies were performed to further sort particles. Best results were obtained when using a regularization parameter of $T = 3$ and 6 classes. The two best classes with the highest resolution containing 286,390 particles were selected. These particles were subjected to another 3D refinement job using a soft mask and resulting in a map at a resolution of 3.3 Å. Despite a slightly lower resolution compared to map obtained form the first 3D refinement, the map quality appeared better upon visual inspection. Additional attempts to reduce structural heterogeneity using local refinements and 3D classifications did not further improve the map quality. Correction of asymmetrical and symmetrical aberrations, and magnification anisotropy of the data set, as well as, re-estimation of the defocus values for each particle were performed. Subsequent particle polishing and 3D refinement yielded a map with a resolution of 2.8 Å (Supplementary Fig. 3). A local resolution-filtered map was generated using a B-factor of −30 Å$^2$ and used for model building. Local resolutions were calculated in ResMap v.1.95[59] and visualized in UCSF Chimera v.1.15[60].

### Model building and refinement

Ab initio model building was carried out in *Coot* v.0.9.6[61] and facilitated by comparing with the predicted structural model generated in AlphaFold2. Model refinement was performed using phenix.real_space_refine in PHENIX v.1.20[62]. The final model was validated using Molprobity[63] and its correlation to the EM map assessed using phenix.mtriage in PHENIX v.1.20[62]. The resulting refinement statistics are summarized in Supplementary Table 2. More than 80% of the residues in the expressed constructs were built, including the residue range aa89-aa728 in EXT1 and aa78-aa702 in EXT2, thus lacking the presumably flexible N- and C-terminal tails and several short, surface exposed loops in EXT1 (aa299-aa310, aa486-aa496, aa515-aa545, and aa678-aa697) and in EXT2 (aa116-aa122, aa287-aa295, aa498-aa505, and aa650-aa670). Several side chains in EXT1 (K111, K267, T273, I275, R311, D313, R314, E318, E320 and K321) and EXT2 (E244, E270, K275, and E278), for which no EM density was observed, were trimmed down to their Cβ. In addition to the EXT1 and EXT2 polypeptide chains, the EM map also contained two *N*-glycans (Asn330 in EXT1 and Asn637 in EXT2) and a UDP molecule in the EXT1 GlcA-T active site. Figures were prepared using UCSF ChimeraX v.1.3[64].

### Generation and validation of knock-out cell lines

HeLa knock-out cell lines were generated using the high-fidelity CRISPR/Cas9 system as desbribed previously[65,66]. Briefly, HeLa CCL-2 cells (ATCC, reference HeLa CCL-2) were transfected with gRNA sequence 5'-AAGTTACCAAAACATTCTAG-3' to target the exon 1 of *EXT1* or the gRNA sequence 5'-GCCTAACAACCGGCACATCA-3' to target exon 2 of *EXT2*. Positively transfected cells were selected with puromycin (3 ng mL$^{-1}$). Knock-out efficiency at the polyclonal level and frameshift confirmation of the individual clones was performed by isolating the cell genomic DNA using the DirectPCR Lysis Reagent

(Viagen Biotech, Los Angeles, USA) and treatment with 0.36 mg mL$^{-1}$ of proteinase K. Targeted loci were amplified by PCR using the following primer pairs 5'-AGTTAAAGAAATCGCCCACATGC-3' (EXT1 forward) and 5'-CTGGTTTCTGTTTAAAGTATCCAGACTCAGGACAAAGAGG-3' (EXT1 reverse), 5'-AGGCCAAAGTGGGAGGATTG-3' (EXT2 forward) and 5'-TGTAAGTGCTACGAGGAGGTG-3' (EXT2 reverse) and analyzed by Sanger sequencing. Analysis of the sequencing reads using the tracking of indels by decomposition (TIDE) software v.3.3.0[67] showed that the *EXT2* knockout cell line contains frameshift mutations corresponding to deletions of −7 and −5 nucleotides and an insertion of +1 nucleotide. For the *EXT1* knockout cell line, we were unable to identify small insertions/deletions (+/−15 nt) using the TIDE software and analysis of PCR products by agarose gel electrophoresis suggested that larger DNA fragments of around 20, 100 and 500 nucleotides have been inserted.

### Complementation assay using knock-out cells

Synthetic genes encoding full-length EXT1 and EXT2 were cloned into the pTWIST CMV BetaGlobin WPRE Neo vector (Twist Bioscience). The EXT1 expression construct contained a N-terminal 3x FLAG and c-myc tag, followed by a 3C protease cleavage site, the EXT2 expression construct encoded a 3x FLAG and Strep II tag, followed by a 3C protease cleavage site at its N-terminus. Point mutations were introduced using site-directed mutagenesis PCR. HeLa CCL-2 WT (ATCC, catalog reference HeLa CCL-2) and knock-out cells ($4 \times 10^5$) were seeded overnight in duplicate wells of a 6-well plate. Cells (at ~70 % of confluence) were transfected, the following day, using jetPRIME (4 μl) and the different DNA constructs (2 μg), according to the manufacturer's recommendations. After 36 h at 37 °C, cells were detached using Versene and counted. $5 \times 10^5$ were resuspended in PBS containing 1 mM EDTA, 1% (w/v) bovine serum albumine, and fixed with 4% paraformaldehyde before immunolabelling[68]. Cell surface HS level was examined after staining with anti-HS antibody (10E4, amsbio. catalog number 370255-1) at 5 μg ml$^{-1}$ and FLAG-tagged enzymes were detected by using anti-FLAG® M2 antibody (Sigma, catalog number F1804), at 2 μg ml$^{-1}$. The cells were then stained with Cy3-conjugated anti-mouse immunoglobulins at a dilution of 1:300 (Jackson ImmunoResearch, catalog number 115-165-068), and analyzed by flow cytometry (Macs Quant v.2.13.0 software, Miltenyi). The fraction of cells displaying HS on their cell surface and cellular FLAG-tagged EXT1 and EXT2 proteins were determined as indicated in Supplementary Fig. 12. The HS content was then normalized by dividing through the fraction of cells expressing EXT1 or EXT2 (FLAG-signal). The average and standard deviation of three independent experiments was determined and a one-sided paired student t-test was performed to calculate the statistical significance of the difference between KO cells transfected with wild-type or mutant EXT1 and EXT2 proteins. The degree of freedom was 2 and t-Stat values were 9.2 for EXT1 D162N/D164N, 15.5 for EXT1 R346, 3.0 for EXT1 D565N/D567N, −1.5 for EXT2 R266A, 7.4 for EXT2 D538N/D540N and −5.2 for EXT2 N637A. Obtained *p*-values are indicated in Fig. 4.

### Mass spectrometry-based proteomic analysis

Purified EXT1-EXT2 was digested in-gel using trypsin (modified, sequencing purity, Promega)[69]. The resulting peptides were analyzed by online nanoliquid chromatography coupled to MS/MS (Ultimate 3000 RSLCnano and Q-Exactive Plus, Thermo Fisher Scientific) using a 50 min gradient. For this purpose, the peptides were sampled on a precolumn (300 μm x 5 mm PepMap C18, Thermo Scientific) and separated in a 75 μm x 250 mm C18 column (Reprosil-Pur 120 C18-AQ, 1.9 μm, Dr. Maisch). The spray voltage was set at 1.5 kV and the heated capillary was adjusted to 250 °C. Survey full-scan MS spectra (m/z = 400–1600) were acquired with a resolution of 70,000 after the accumulation of $1 \times 10^6$ ions (maximum filling time 250 ms). Up to 10 most intense ions were then individually selected for fragmentation by

higher-energy collisional dissociation (normalized collision energy: 30%). MS/MS spectra were then acquired with a resolution of 17,500 after the accumulation of $10^6$ ions (maximum filling time: 250 ms). The MS and MS/MS data were acquired by Xcalibur v4.0 (Thermo Fisher Scientific). The experiment was carried out once ($n = 1$). As purified protein was analyzed no controls were included.

Peptides and proteins were identified by Mascot (version 2.8.0, Matrix Science) through concomitant searches against the Uniprot database (*Homo sapiens* taxonomy, 20220114 download), a homemade database containing the sequences of TEV-cleaved EXT1 and EXT2 proteins, and a homemade database containing the sequences of classical contaminant proteins found in proteomic analyses (bovine albumin, keratins, trypsin, etc.). Trypsin/P was chosen as the enzyme and two missed cleavages were allowed. Precursor and fragment mass error tolerances were set at 10 and 20 ppm, respectively. Peptide modifications allowed during the search were: Carbamidomethyl (C, fixed), Acetyl (Protein N-term, variable), and Oxidation (M, variable). The Proline software[70] was used for the compilation, grouping, and filtering of the results (conservation of rank 1 peptides, peptide length ≥ 6 amino acids, false discovery rate of peptide-spectrum-match identifications <1%, and minimum of one specific peptide per identified protein group). Proline was then used to perform a MS1 quantification of the identified protein groups. Intensity-based absolute quantification (iBAQ) values were calculated for each protein from MS1 intensities of razor and unique peptides. Proteins identified in the contaminant database and additional keratins were discarded from the list of identified proteins.

## Mass photometry analysis
Mass photometry experiments were carried out using a Refeyn One$^{MP}$ instrument (Oxford, UK). Microscope coverslips were cleaned several times with $H_2O$ and isopropanol and then air-dried. A silicone gasket was placed onto the clean coverslip and 19 μL of buffer B were applied. The focal position was identified automatically using the Refeyn Acquire$^{MP}$ v.2.4.0 software. 1 μL of purified EXT1-EXT2 at a concentration of 600 nM were added to the buffer drop, resulting in a final protein concentration of 30 nM. Movies with a duration of 60 s, corresponding to 6000 frames, were recorded using a regular field-of-view acquisition setting. Corresponding mass for automatically detected particles was calculated using the Refeyn Discover$^{MP}$ v.2.5.0 software. Mass kernel density were estimated with a 4.6 kDa bandwidth. Contrast-to-mass calibration was performed using a NativeMark$^{TM}$ unstained protein standard (Thermo Fisher Scientific), containing proteins with a molecular weight of 66, 146, 480, and 1048 kDa.

## Nano differential scanning fluorimetry
The thermal stability of purified EXT1-EXT2 wild-type and mutant complexes was measured using nano-differential scanning fluorimetry. Around 12 μL of protein complex at a concentration of 0.15 mg mL$^{-1}$ in buffer B were applied into standard-grade NanoDSF glass capillaries (Nanotemper). Capillaries were loaded into a Prometheus NT.48 device (Nanotemper) and excitation power was set to 80% using the PR.ThermControl v.2.3.1 software. Samples were heated from 15 °C to 95 °C using a temperature slope of 2 °C per minute. Melting points of duplicate measurements were determined using the PR.ThermControl v.2.3.1 software.

## Negative stain analysis
Purified wild-type and EXT1 D162N/D164N containing EXT1-EXT2 was diluted to 0.02 mg mL$^{-1}$ in buffer A. Grids were prepared using the Negative Stain-Mica-carbon Flotation Technique (MFT) - Valentine procedure. In short, samples were absorbed to the clean side of a carbon film on mica, stained, and transferred to a 400-mesh copper grid. The images were taken under low dose conditions (<10 e$^-$ Å$^{-2}$) with defocus values between 1.2 and 2.5 μm on a Tecnai 12 LaB6

electron microscope at 120 kV accelerating voltage using a CCD Camera Gatan Orius 1000.

## Reproducibility
Protein purifications of EXT1-EXT2 WT complexes have been carried out more than ten times resulting in similar purities as in the exemplary SDS-PAGE image shown in Fig. 1a. Purification of mutant EXT1-EXT2 complexes as shown in Supplementary Fig. 11a was performed only once. In vitro activity assays were carried out in independent triplicates providing similar results as shown in Figs. 1c, 3f, 4b, and 5b. Representative electron micrographs were chosen from a set of more than 5800 cryo-EM images (Supplementary Fig. 3b), 20 negative stain images for EXT1-EXT2 WT complexes (Supplementary Fig. 11d), and 10 negative stain images for EXT1 D162N/D164N containing EXT1-EXT2 complexes (Supplementary Fig. 11e).

## Reporting summary
Further information on research design is available in the Nature Portfolio Reporting Summary linked to this article.

## Data availability
The atomic structure coordinates of the human hetero-dimeric complex EXT1-EXT2 were deposited in the Protein Data Bank under the accession number 7ZAY. The corresponding cryo-EM maps were deposited in the Electron Microscopy Data Bank under the accession number EMD-14582. The structures of EXTL2, EXTL3, and POGLUT1 used in this study are available in the Protein Data Bank under the accession numbers 1ON6, 7AUA, and 5L0U. The mass spectrometry proteomics data have been deposited to the ProteomeXchange Consortium under the accession number PXD034871. Source data are provided with this paper.

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

## Acknowledgements

We thank Rida Awad, Farah Fouladkar, Damien Maurin, and Yoan R. Monneau for their contribution in carrying out molecular biology experiments. We thank Evelyne Gout and Mélanie Friedel for help with cellular studies. We thank Caroline Mas for assistance to perform mass photometry analysis, Michel Thepaut for help during nano differential scanning fluorimetry experiments, Annie Adrait for help during mass spectrometry-based proteomic analysis, Jérôme Hénault for help during oligosaccharide substrate analog synthesis and Jean-Philippe Kleman for help during flow cytometry analysis. This research was funded by the Agence Nationale de la Recherche, grant number [ANR-18-CE11-0006-01] and [ANR-10-INBS-08], the ATIP-Avenir programm and the "Investissements d'avenir" program Glyco@Alps, grant number [ANR-15-IDEX-02]. We acknowledge the European Synchrotron Radiation Facility for provision of beam time on CM01 (https://doi.org/10.15151/ESRF-ES-599588899). This work used the EM facility, the cell imaging platform and the biophysics characterization platform of the Grenoble Instruct-ERIC Center (ISBG; UMS 3518 CNRS CEA-UGA-EMBL) with support from the French Infrastructure for Integrated Structural Biology (FRISBI; ANR-10-INSB-05-02) and GRAL, a project of the University Grenoble Alpes graduate school (Ecoles Universitaires de Recherche) CBH-EUR-GS (ANR-17-EURE-0003) within the Grenoble Partnership for Structural Biology. The IBS Electron Microscope facility is supported by the Auvergne Rhône-Alpes Region, the Fonds Feder, the Fondation pour la Recherche Médicale and GIS-IbiSA.

## Author contributions

R.W. and H.L.-J. designed the project. J.O. and R.W. purified EXT1-EXT2 wild-type and mutant complexes and performed in vitro glycosylation assays, mass photometry and nano diffrential scanning fluorimetry experiments. Y.C. performed mass spectrometry experiments. D.F. carried out negative stain EM analysis. C.L.N and D.B. synthesized fluorescent substrate analog and confirmed its quality. R.W. carried out enzymatic extension of substrate analog. R.W. prepared and screened samples for cryo-EM experiments. M.H., G.S., and R.W. performed cryo-EM data collection. F.L. and R.W. processed cryo-EM data. F.L. built the EXT1-EXT2 model and carried out model refinement. F.L. and R.W. analyzed the structure. J.B. generated and validated knock-out cell lines. R.S. performed *in cellulo* mutational analysis. J.O. analyzed patient mutations. F.L., J.O., R.W. and H.L.-J. wrote the manuscript and all authors contributed to its revision.

## Competing interests

The authors declare no competing interests.
