## [Peer Review File · Nature Communications]

REVIEWER COMMENTS

Reviewer #1 (Remarks to the Author):

Leicisco et al present a cryo-EM structure of the EXT1-EXT2 heparan sulfate polymerizing enzyme. This is the bifunctional glycosyltransferase responsible for elongating the growing HS chain during biosynthesis, and is a key player in production of this essential biological polymer. Despite progress in many HS biosynthesis enzyme structures being solved over the years, experimental determination of EXT1-EXT2 has so far been elusive. Here, the authors have expressed recombinant EXT1-EXT2, solved its structure, and presented some preliminary structure function analyses. The data confirm that EXT1-EXT2 is an obligate heterodimer and that functional GlcNAcT domains are present on both subunits, but EXT2 lacks a GlcAT domain. Finally, a model is proposed suggesting distributive polymerization of HS by the enzyme. Given the importance of HS polysaccharides, the EXT1-EXT2 enzyme is of high interest to the glycobiology community. Indeed, I should declare we looked at this enzyme ourselves a while ago, but gave up, having made nowhere near as much progress as this team. The work presented is technically excellent and well written, as I would expect from these authors.

On the other hand, I am slightly disappointed by the level of insight that has been presented here. The great power of structural biology is the ability derive functional details about proteins from molecular views of them in action. In this regard, a single structure of EXT1-EXT2, with a UDP in only one of the active sites, does not shed a huge amount of light on the mechanistic details of activity. I gather from the experimental write-up that the solved structure was very similar to the AlphaFold prediction. A difficult point might be that many of the functional insights presented here (e.g. the inactive EXT2 GlcA-T domain, mapping of patient mutations) could have been inferred from the predicted structure alone.

To be clear, I am **not** suggesting that experimental structures are not worth publishing unless they are novel compared to AF predictions. Experimental confirmation of predicted data is always valuable. However, given the insights that incredibly accurate AF models now provide (<https://doi.org/10.1101/2021.09.26.461876>), I do think a bit more is needed to reach the novelty bar for this journal.

- What I think would really lift this paper is analysis of more substrate complex structures. In particular, I would love to see the authors try for a structure of EXT1-EXT2 with a HS oligosaccharide acceptor (one of the oligos used for biochemical assays might work well), or perhaps a linker tetrasacch. Having high-quality structures with acceptor substrates would deliver substantial insights into EXT1-EXT2 function, and start to speak towards the molecular details of catalysis. Given the EM

resources available to this team, I do not think a few additional dataset collections should be overly demanding.

- Regarding UDP-sugar complexes. Might some of the other models from hetero refinement be complexed with UDP or UDP-sugar in other subsites? Substrate binding to glycosyltransferase domains can often induce substantial conformational movements – perhaps these are in the other classes? Were they refined further?

- In general, were there regions in the experimental structure that were substantially different from the predicted model? Do these have any relevance for activity? Loops, interface regions etc?

Additional minor points

- P5-L164 and others – ‘EXT-like 2’. The gene name is EXTL2, which I realise does stand for EXT-like 2. However, throughout, the proper gene name should be used.

- The EXT2 ‘GlcA-T domain’ is referenced throughout the paper, even though one of the central claims presented is that this ‘domain’ is inactive. Perhaps better to refer to it as a ‘pseudo-GlcA-T’ domain, or a ‘GlcA-T like’ domain.

- P7 L248 – ‘The Achilles heel of the EXT1-EXT2 complex’ – I’m not sure what this phrase means. It seems overly anthropomorphic and would be better removed.

- P8 L290 – ‘Recombinant EXT1-EXT2 can be exploited to generate oligosaccharides of defined length, such as the nona-saccharide described in this work, a missing tool for studying the specific interaction between proteins and HS’.

Whilst it is technically true EXT1-EXT2 can be used for making HS oligos, the capability to generate defined HS has been around for a while, thanks to bacterial glycosyltransferases such as KfiA, KfiC and pmHS2. I certainly would not call EXT1-EXT2 a ‘missing tool’ unless the authors can demonstrate that it is better than existing tools for this application. If not, I suggest this sentence is removed.

Reviewer #2 (Remarks to the Author):

In this important paper, the structure of the human heparan sulfate (HS) polymerase complex EXT1-EXT2 is reported. The results presented sheds light on the process of HS elongation and is a very welcome contribution to the field. The technical quality of the report is high and the manuscript is carefully written with good illustrations. Some minor issues can be discussed including the interpretation of some experiments.

1. Staining with the antibody 10E4 depends not only on the amounts of HS but also of HS structure, which may have affected the results presented in Fig. 3. Did you investigate the structure of HS after transfection of the knockout cells? Expression of the EXT proteins have previously been shown to affect N-sulfation which in turn influences further modification.

2. In Fig. 3b, the enzyme activities of EXT1-M2 and EXT2-M2 are missing.

3. It is difficult to see how the results presented in Fig. 3c match the flow-cytometry data in Suppl. Fig. 10.

4. The human gene names of EXT1 and EXT2 are EXT1 and EXT2.

5. Regarding the processivity, I am not convinced that the results exclude the possibility that elongation occurs in a processive manner. Maybe the complex will have another conformation when bound to a natural substrate. Substrate-binding could also possibly affect the structure of the complex, maybe bringing the GlcA-T domain closer to one or both of the GlcNAc-T domains. The model proposed in Fig. 4 is not clear to me. Do you mean that two HS chains are being elongated at the same time?

Reviewer #3 (Remarks to the Author):

The manuscript by Francisco Leisico et al. describes the structure and function of the human heparan sulfate polymerase complex EXT1-EXT2. The enzymes are bifunctional glycosyltransferases and generate the glycan backbone of heparan sulfates, consisting of alternating glucuronic acid and N-acetylglucosamine units. The authors present cryo EM structures of a recombinantly expressed EXT1/2 complex combined with functional analyses to support a model by which EXT1 and EXT2 differ substantially in catalytic activity.

As it stands, the observations are interesting yet do not provide substantial insights into the reaction mechanisms of the individual transferases. Accordingly, the work falls short in providing mechanistic insights into heparan sulfate biosynthesis. Major revisions seem to be necessary to warrant publication.

Major points:

Structural analyses:

Fig. 2b, UDP map quality: The UDP map shown in Fig. 2b appears to be at a very low contour level and does not fit the modeled UDP well. Please show a larger region of the EM map including surrounding backbone and side chain volumes to relate the UDP map quality to the surrounding protein. Also, please state the contour level in the caption.

Mutagenesis analyses: None of the EXT2 point mutations are actually located in the potential active site, therefore, their relevance for function is questionable. For example, it is unclear why the authors chose D139 and D141 for mutagenesis of the EXT2 GlcA-T domain (EXT2-M1). These residues are far away from the putative substrate binding site and hence unlikely to impact catalysis. To avoid confusions with, for example, family 2 GTs, it is perhaps best not to refer to this motif as a DxD motif as this implies the residues are implicated in nucleotide/cation coordination. The authors are encouraged to review representative GT-B ligand complexes and reaction mechanisms (for example SN1 versus SN2). Perhaps other residues can be identified that impact ligand binding? This is an important point because the mutagenesis analyses affect the proposed biological interpretation.

Fig. 3b and c: The effects of point mutations shown in panels b and c are inconsistent. For example, EXT1-M3 is inactive in panel B, yet has almost wild type activity in panel c. While EXT1-M3 is completely inactive (in panel b), the EXT2-M3 mutation reveals about 50% residual activity. How can this be explained? Does EXT2 require a functional EXT1 for activity. The protein expression protocols and in particular the location of affinity tags differ for complexes used in panels b and c. Is it possible that C-terminal tags interfere with GlcNAc-T activity?

Substrate binding to EXT2 GlcA-T, EDF 9: The text states that substrate binding to the domain is likely prevented by 'an anti-parallel beta-sheet intruding into the donor substrate binding pocket'. Based on all figures showing this hairpin, it seems to be pointing away from the putative binding site. Hence, this argument should be revised or explained in more detail.

Proposed reaction mechanism: All proposed reaction mechanisms (SN2 versus SN1) require knowledge of the acceptor binding sites as well as the base catalyst (at least for SN2). The manuscript fails to discuss any mode of acceptor binding, although the authors synthesized the dp8 and dp9 oligosaccharides that serve as acceptors in vitro. Insights into acceptor binding are required to validate any mechanistic insights.

Minor points:

The manuscript lacks discussions of previous data on EXT1 and EXT2 purification and function, showing that both enzymes can indeed be purified separately. Further, previous data shows that the EXT1/EXT2 heterodimer has much higher GlcA-T activity than it does GlcNAc-T activity (McCormick et al, 2000; Busse & Kusche-Gullberg, 2003). This is not discussed at all.

EXD 5c and d: Please state what glycans have been modeled.

EXD 6b and d: Please define the distances of the interactions shown as dashed lines.

Fig. 3C: The Y-axis requires a label (normalized abundance?).

FACE electrophoresis: FACE was invented in the 90s. This should be acknowledged.

Figure 4: As shown the figure is incorrect as it misses the tetrasaccharide linker region.

Article “Structure of the human heparan sulfate polymerase complex EXT1-EXT2”

Answer to Reviewers

Reviewer #1 (Remarks to the Author):

Leicisco et al present a cryo-EM structure of the EXT1-EXT2 heparan sulfate polymerizing enzyme. This is the bifunctional glycosyltransferase responsible for elongating the growing HS chain during biosynthesis, and is a key player in production of this essential biological polymer. Despite progress in many HS biosynthesis enzyme structures being solved over the years, experimental determination of EXT1-EXT2 has so far been elusive. Here, the authors have expressed recombinant EXT1-EXT2, solved its structure, and presented some preliminary structure function analyses. The data confirm that EXT1-EXT2 is an obligate heterodimer and that functional GlcNAcT domains are present on both subunits, but EXT2 lacks a GlcAT domain. Finally, a model is proposed suggesting distributive polymerization of HS by the enzyme. Given the importance of HS polysaccharides, the EXT1-EXT2 enzyme is of high interest to the glycobiology community. Indeed, I should declare we looked at this enzyme ourselves a while ago, but gave up, having made nowhere near as much progress as this team. The work presented is technically excellent and well written, as I would expect from these authors.

On the other hand, I am slightly disappointed by the level of insight that has been presented here. The great power of structural biology is the ability derive functional details about proteins from molecular views of them in action. In this regard, a single structure of EXT1-EXT2, with a UDP in only one of the active sites, does not shed a huge amount of light on the mechanistic details of activity. I gather from the experimental write-up that the solved structure was very similar to the AlphaFold prediction. A difficult point might be that many of the functional insights presented here (e.g. the inactive EXT2 GlcA-T domain, mapping of patient mutations) could have been inferred from the predicted structure alone.

To be clear, I am **not** suggesting that experimental structures are not worth publishing unless they are novel compared to AF predictions. Experimental confirmation of predicted data is always valuable. However, given the insights that incredibly accurate AF models now provide (<https://doi.org/10.1101/2021.09.26.461876>), I do think a bit more is needed to reach the novelty bar for this journal.

- What I think would really lift this paper is analysis of more substrate complex structures. In particular, I would love to see the authors try for a structure of EXT1-EXT2 with a HS oligosaccharide acceptor (one of the oligos used for biochemical assays might work well), or perhaps a linker tetrasacch. Having high-quality structures with acceptor substrates would deliver substantial insights into EXT1-EXT2 function, and start to speak towards the molecular details of catalysis. Given the EM resources available to this team, I do not think a few additional dataset collections should be overly demanding.

Our answer: We thank reviewer #1 for the comments and we agree that acceptor bound structures would provide a very valuable insight. Solving such a structure complex would require determining precise binding affinities towards different substrates to predict the best sample preparation conditions. Based on the technical challenges in expressing and purifying the hetero-dimeric EXT1-EXT2 complex, this would be extremely difficult to obtain and we currently don't have the tools to develop such a project. We have, despite of these challenges, prepared a new cryo-EM sample, in which the EXT1-EXT2 complex was mixed with two acceptor oligosaccharides. We chose a commercial dp7 and dp8 acceptor substrate (IDURON, UK), featuring GlcA and GlcNAc at the non-reducing end, respectively. These oligosaccharides have the advantage, that they only harbor a

p-nitrophenyl group at the reducing end rather than a large fluorophore present in the substrates used for the activity assays. Also, we do not have sufficient quantities of chemo-enzymatically synthesized dp9 oligosaccharide for structural studies. The final concentration of each of the acceptor oligosaccharides was 100 μ M (the highest possible concentration we could reach with our stocks). $MnCl_2$ was added at a final concentration of 5 mM. No UDP-sugar donors were added to this sample. We incubated this mixture for 30 min on ice and then prepared cryo-EM grids. We collected a data set containing 10929 images on a 300 kV Titan Krios TEM (at ESRF, Grenoble) and obtained a 3D reconstruction at a nominal resolution of 3.2 \AA with a highly similar overall architecture compared to our UDP-bound structure (Fig. 1 below). Despite adding the acceptor substrates for the GlcNAc-T and GlcA-T reaction, we did not observe any additional densities in any of the four catalytic sites. A reason for this could be that acceptor substrate binding occurs only transiently or that the binding affinity of the used acceptor substrates is very low. The only difference between the two cryo-EM maps was observed in the active site of the EXT1-GlcA-T domain, in which no density for a UDP ligand was found. Our additional attempt thus resulted in an apo structure. We do not include this new structure into our manuscript as it does not provide additional mechanistic insight.

Fig. 1 | Cryo-EM structure determination of EXT1-EXT2 in presence of acceptor substrates. **a**, Representative micrograph. **b**, Selected 2D classes. **c**, Comparison of EXT1-EXT2 cryo-EM maps. On the left, the UDP-bound structure described in the first version of the manuscript. On the right side, the new cryo-EM map is shown. Despite the addition of acceptor substrates dp7 and dp8, the complex is found to be in an apo state. **d**, Close-up view onto the GlcA-T active site in EXT1. Apo and UDP-bound structures were superimposed to illustrate the UDP ligand in the GlcA-T active site of the apo structure. Only the UDP-bound structure features density for the UDP ligand.

In order to provide additional mechanistic insights, we re-examined our cryo-EM structure and performed additional structural comparisons. When studying the GlcA-T catalytic site, we found aspartate residues Asp162 and Asp164 that could act as a base catalyst and arginine residues Arg280 and Arg346 which could stabilize the “pyrophosphate” leaving group of UDP. Based on these findings, we propose, that GlcA-T transfer follows a S_N2 -like mechanism.

An additional paragraph describing the mechanistic insights provided by our cryo-EM structure has been added to the “Proposed mechanism for chain polymerization” section:

“The high structural similarities of the GlcNAc-T active sites of EXT1 and EXT2 with EXTL2, support a S_Ni -like reaction mechanism for GlcNAc transfer. Currently, there are no structural homologs of the GlcA-T domain of EXT1 and EXT2 known. Nevertheless, we propose that GlcA transfer is based on a S_N2 -like reaction mechanism. An important feature of S_N2 -like inverting GTs is the presence of a base catalyst, which deprotonates and thus activates the C4-hydroxyl group of the acceptor substrate for its nucleophilic attack on the C1 carbon atom of the donor substrate⁴⁰. Structural comparison of EXT1 GlcA-T and POGLUT1 suggests that either Asp162 or Asp164 might play this role (Supplementary Fig. 7). The arginine residues Arg280 and Arg346 are in close proximity to the pyrophosphate moiety of the UDP and are likely to stabilize the negative charge of the leaving group. Together, with the spatial constraints of the donor and acceptor binding pocket, our observations favor a S_N2 -like mechanism.”

- Regarding UDP-sugar complexes. Might some of the other models from hetero refinement be complexed with UDP or UDP-sugar in other subsites? Substrate binding to glycosyltransferase domains can often induce substantial conformational movements – perhaps these are in the other classes? Were they refined further?

Our answer: We thank reviewer #1 for pointing this out. We have performed multiple data processing strategies to analyze conformational heterogeneity or the presence of UDP-sugar substrates in sub-stoichiometric amounts. More precisely, we have performed the following experiments: i) 3D classification using different values for the regularization parameter T and varying numbers of classes in Relion3.1 ii) 3D classification without alignment in Relion3.1 iii) 3D classification using masks covering either the full complex or smaller regions thereof in Relion3.1 iv) heterogeneous refinement in cryoSPARCv3.3 v) 3D variability analysis and subsequent heterogeneous refinement in cryoSPARCv3.3.

All promising reconstructions were subsequently refined using auto-refinement. We studied the 4 active sites in all the obtained 3D reconstructions and consistently found a single UDP molecule bound in the GlcA-T active site of EXT1. The 3D classification job described in the manuscript provided the best overall electron density map and was thus chosen for further processing steps.

The following sentence was added to the method section:

“Different 3D classification strategies were performed to further sort particles. Best results were obtained using a regularization parameter of $T=3$ and 6 classes.”

- In general, were there regions in the experimental structure that were substantially different from the predicted model? Do these have any relevance for activity? Loops, interface regions etc?

Our answer: A new figure (Supplementary Fig. 5) has been added to the manuscript. We first compared our cryo-EM structure with the AlphaFold2 predicted models of EXT1 and EXT2 accessible through the AlphaFold data base. The most prominent difference is the relative orientation of the GlcA-T towards the GlcNAc-T domain of EXT2. Interestingly, when predicting a model for the hetero-dimeric EXT1-EXT2 complex using ColabFold, the EXT2 spatial organization changes and is now in agreement with our cryo-EM structure. Comparing the predicted model of the full complex with the cryo-EM structure, we observe a small rotation or shift of the EXT2 GlcA-T domain in relation to the other domains of the complex.

We have added the following paragraph in the “Architecture of EXT1-EXT2” section:

“Comparison of our experimental structure with the AlphaFold2 models of the individual EXT1 and EXT2 proteins shows that the domain folds are highly similar. Differences include the re-arrangement of the EXT2 GlcNA-T and GlcA-T domains in respect to each other and the presence of an anti-parallel beta-sheet formed between the C-termini of EXT1 and EXT2. These structural

features are correctly predicted when generating a model for the hetero-dimeric complex. (Supplementary Fig. 5).”

We would also like to mention that our cryo-EM provides for the first time experimental evidence for the existence of two *N*-linked glycans, EXT1 N330 and EXT2 N637. The latter appears to be highly ordered and forms a Pi-stacking interaction with Trp437 (see Supplementary Fig. 4). Importantly, we have tried to express and purify a EXT1-EXT2 complex harboring a N637A substitution in EXT2, but the protein was not expressed in a soluble form, suggesting that this *N*-glycan plays a role in complex stability.

Additional minor points

- P5-L164 and others – ‘EXT-like 2’. The gene name is EXTL2, which I realise does stand for EXT-like 2. However, throughout, the proper gene name should be used.

Our answer: Modified as requested.

- The EXT2 ‘GlcA-T domain’ is referenced throughout the paper, even though one of the central claims presented is that this ‘domain’ is inactive. Perhaps better to refer to it as a ‘pseudo-GlcA-T’ domain, or a ‘GlcA-T like’ domain.

Our answer: We thank reviewer #1 for this very useful advice. We have added the following sentence in the “Substrate recognition in the four active sites” section: “*These structural differences suggest that the GlcA-T domain of EXT2 might not be able to carry out glucuronic acid transfer, hereafter referred to as pseudo-GlcA-T domain.*” Figures and text following this statement were modified accordingly.

- P7 L248 – ‘The Achilles heel of the EXT1-EXT2 complex’ – I’m not sure what this phrase means. It seems overly anthropomorphic and would be better removed.

Our answer: Modified as requested.

- P8 L290 – ‘Recombinant EXT1-EXT2 can be exploited to generate oligosaccharides of defined length, such as the nona-saccharide described in this work, a missing tool for studying the specific interaction between proteins and HS’.

Whilst it is technically true EXT1-EXT2 can be used for making HS oligos, the capability to generate defined HS has been around for a while, thanks to bacterial glycosyltransferases such as KfiA, KfiC and pmHS2. I certainly would not call EXT1-EXT2 a ‘missing tool’ unless the authors can demonstrate that it is better than existing tools for this application. If not, I suggest this sentence is removed.

Our answer: We have removed the expression “missing tool” as suggested.

Reviewer #2 (Remarks to the Author):

In this important paper, the structure of the human heparan sulfate (HS) polymerase complex EXT1-EXT2 is reported. The results presented sheds light on the process of HS elongation and is a very welcome contribution to the field. The technical quality of the report is high and the manuscript is carefully written with good illustrations. Some minor issues can be discussed including the interpretation of some experiments.

1. Staining with the antibody 10E4 depends not only on the amounts of HS but also of HS structure, which may have affected the results presented in Fig. 3. Did you investigate the structure of HS after transfection of the knockout cells? Expression of the EXT proteins have previously been shown to affect N-sulfation which in turn influences further modification.

Our answer: We thank reviewer #2 for pointing this out. In order to check that the observed changes in HS levels, when comparing WT HeLa cells with EXT1 and EXT2 KO cell lines and upon complementation with WT or mutant EXT1 and EXT2 constructs, are not due to a change in HS structure, we performed additional FACS experiments.

In addition to measuring HS levels using the 10E4 (Amsbio) antibody, we performed control experiments using the JM403 (Amsbio) antibody. According to the manufacturer's description, these two antibodies recognize different HS epitopes. The FACS experiment using the JM403 reveals a similar restoration of HS levels in complemented EXT1 and EXT2 KO cell lines as compared to the experiments using the 10E4 antibody (Figure 2 below and Figure 4 in the manuscript). The observed difference in HS are thus likely caused by a decrease of HS on the cell surface rather than from a change in HS structure.

Fig. 2 | Quantification of cell surface HS levels using the JM403 antibody. Flow cytometry experiments were performed using HeLa knock-out (KO) cell lines lacking either the EXT1 or EXT2 protein. The knock-out cell lines were not transfected or complemented with either WT EXT1 or EXT2 proteins. Quantified HS content from three independent experiments was normalized to the EXT expression level (as described in the method section of the manuscript) and blotted as percentage compared to wild-type HeLa cells (n=3), error bars show standard deviation.

2. In Fig. 3b, the enzyme activities of EXT1-M2 and EXT2-M2 are missing.

Our answer: We have now generated EXT1-R346A and EXT2-R266A protein expression constructs and we have expressed, purified and analyzed the stability of these complexes by nano differential scanning fluorimetry. We studied the in vitro activity of these mutant complexes by FACE.

The results were added to Fig. 3 and Supplementary Fig. 9. They are described in the main text alongside with the other mutants. We have renamed mutants in the manuscript for easier understanding.

3. It is difficult to see how the results presented in Fig. 3c match the flow-cytometry data in Suppl. Fig. 10.

Our answer: The figure legend has been modified to describe performed experiments and their analysis in a clearer manner.

4. The human gene names of EXT1 and EXT2 are EXT1 and EXT2.

Our answer: We were not fully sure what reviewer #2 tried to suggest, but we changed the gene names from “*ext*” (small, italics) to “*EXT*” (capitals, italics).

5. Regarding the processivity, I am not convinced that the results exclude the possibility that elongation occurs in a processive manner. Maybe the complex will have another conformation when bound to a natural substrate. Substrate-binding could also possibly affect the structure of the complex, maybe bringing the GlcA-T domain closer to one or both of the GlcNAc-T domains. The model proposed in Fig. 4 is not clear to me. Do you mean that two HS chains are being elongated at the same time?

Our answer: We thank reviewer #2 for this comment. In order to strengthen our hypothesis that chain elongation is a nonprocessive process, we have performed a time course experiment during which we followed the chain polymerization reaction in vitro. The results are illustrated in Figure 5. A corresponding description of the experiment has been added to the method section and the description of the chain elongation mechanism has been rephrased in the “Proposed mechanism for chain polymerization” section:

“Interestingly, the EXT1-EXT2 architecture suggests that HS chain elongation is a nonprocessive reaction as its active sites are located on opposite sites of the complex and, in addition, are facing away of each other (Fig. 5b). Such a spatial arrangement, with a distance of 50Å between the active sites, is in line with a distributive, multi-step process, during which the elongated polysaccharide chain detaches from EXT1-EXT2 before entering into the next catalytic site. A time course experiment following the chain elongation in vitro further supports this hypothesis as we observed the formation of oligosaccharides of various lengths, rather than the direct conversion of the oligosaccharide substrate to a fully extended HS chain, as it would be expected in case of a processive chain elongation (Fig. 5a)⁴¹. ”

Reviewer #3 (Remarks to the Author):

The manuscript by Francisco Leisico et al. describes the structure and function of the human heparan sulfate polymerase complex EXT1-EXT2. The enzymes are bifunctional glycosyltransferases and generate the glycan backbone of heparan sulfates, consisting of alternating glucuronic acid and N-acetylglucosamine units. The authors present cryo EM structures of a recombinantly expressed EXT1/2 complex combined with functional analyses to support a model by which EXT1 and EXT2 differ substantially in catalytic activity. As it stands, the observations are interesting yet do not provide substantial insights into the reaction mechanisms of the individual transferases. Accordingly, the work falls short in providing mechanistic insights into heparan sulfate biosynthesis. Major revisions seem to be necessary to warrant publication.

Major points:

Structural analyses:

Fig. 2b, UDP map quality: The UDP map shown in Fig. 2b appears to be at a very low contour level and does not fit the modeled UDP well. Please show a larger region of the EM map including surrounding backbone and side chain volumes to relate the UDP map quality to the surrounding protein. Also, please state the contour level in the caption.

Our answer: We thank reviewer #3 for pointing out the importance of comparing the density for the ligand with the one surrounding the protein residues. We have replaced the UDP map shown in Fig. 2b with a larger image showing the EM map not only around the UDP, but also around the backbone and the side chains, as suggested. Supplementary Fig. 6 has been added.

Of note, examination of the UDP's furanose moieties in 16 pdb structures of glycosyl transferases with GT-B fold (1bgu, 1c3j, 3pe3, 3pe4, 4ay5, 4gyw, 4gz3, 4whm, 5l0r, 5l0s, 5l0t, 4w6q, 7erx, 7es1, 7es2 and 7fg9) shows that the sugar rings adopt either the ²E or the ³T₄ conformations. Regarding the five-member ring atoms, the major difference between those two conformations is the position of the C3 and O3 atoms. This can explain the lower electron density at this position in the UDP map shown in the previous Fig. 2 when assuming an equilibrium between the ²E and ³T₄ conformations within the GlcAT site of EXT1.

We have added the following sentence in the "Substrate recognition in the four active sites" section: "Examination of GT-B fold glycosyl transferase structures shows that UDP furanose rings adopt either the ²E or the ³T₄ conformation. These two conformations differ mainly in the position of the C3 and O3 atoms, which could explain the lower electron density in this part of the EM map."

Mutagenesis analyses: None of the EXT2 point mutations are actually located in the potential active site, therefore, their relevance for function is questionable. For example, it is unclear why the authors chose D139 and D141 for mutagenesis of the EXT2 GlcA-T domain (EXT2-M1). These residues are far away from the putative substrate binding site and hence unlikely to impact catalysis. To avoid confusions with, for example, family 2 GTs, it is perhaps best not to refer to this motif as a DxD motif as this implies the residues are implicated in nucleotide/cation coordination. The authors are encouraged to review representative GT-B ligand complexes and reaction mechanisms (for example SN1 versus SN2). Perhaps other residues can be identified that impact ligand binding? This is an important point because the mutagenesis analyses affect the proposed biological interpretation.

Our answer: We thank reviewer #3 for his/her comments:

1) We had a closer look onto all residues located in the active sites of EXT1 GlcA-T and EXT2 pseudo-GlcA-T and examined their potential involvement into the catalytic mechanism.

Three important residues for UDP binding are found in EXT1: Y319, R346 and R280. Only the R266 in EXT2, corresponding to R280 in EXT1, is conserved. The importance of residue R280A is supported by a structural comparison with the UDP binding site of POGLUT1, another GT-B fold glycosyltransferase. Our EXT1 cryo-EM structure and the POGLUT1 crystal structure suggest that the arginines R280 and R218, respectively, are involved in binding the pyrophosphate moiety of the UDP, thereby stabilizing this leaving group.

We generated two new constructs to express and purify EXT1-EXT2 complexes harboring mutations in the GlcA-T site: EXT1 R280A and EXT2 R266A. Protein stability was measured by nanoDSF and the complexes harboring the EXT1 R346A and EXT2 R266A were found to have a slightly reduced thermal stability. Importantly, the EXT1 R280A mutant does not show in vitro GlcA-T activity, while mutating EXT2 R266A does not alter complex activity. Since the two arginine residues EXT1 R280A and EXT2 R266A are sequence conserved and are located in similar positions in the active sites, our functional data support the hypothesis that only EXT1 has GlcA-T activity.

We have further cloned and expressed two additional complexes containing either the mutation EXT1 Y324A or EXT2 Y308A, as this tyrosine in EXT1 forms a Pi-stacking interaction with the uracil ring of UDP. The proteins were not stably expressed, preventing further functional characterization.

Finally, the aspartate residues D162 and D164 important for EXT1 GlcA-T activity, are already “mutated” to asparagine residues in EXT2 and thus cannot be targeted in our mutational analysis.

To conclude, we have intensively studied all residues located in the pseudo-GlcA-T active site of EXT2.

The newly generated data was added to Fig. 3, Supplementary Fig. 9, Supplementary Fig. 10 and described in the text alongside the other mutants. Mutants were renamed to facilitate comprehension.

2) Concerning the EXT2-D139N/D141N mutant: The EXT2-M1 mutant was removed from the manuscript to avoid confusion, including Supplementary Fig. 9 in first version of the manuscript.

3) We removed the expression “DxD motif” for the GlcA-T domain.

Fig. 3b and c: The effects of point mutations shown in panels b and c are inconsistent. For example, EXT1-M3 is inactive in panel B, yet has almost wild type activity in panel c. While EXT1-M3 is completely inactive (in panel b), the EXT2-M3 mutation reveals about 50% residual activity. How can this be explained? Does EXT2 require a functional EXT1 for activity. The protein expression protocols and in particular the location of affinity tags differ for complexes used in panels b and c. Is it possible that C-terminal tags interfere with GlcNAc-T activity?

Our answer: We thank the reviewer for pointing this out. The strongest effect of mutations for in vitro and in cellulo experiments are observed for mutations D162N/D164N and R346A in the EXT1 GlcA-T active site. The mutations almost fully abolish enzyme activity, also due to the fact that the pseudo-GlcA-T domain of EXT2 does not contribute to GlcA-T activity of the complex.

Mutations in either of the GlcNAc-T domains of EXT1 or EXT2 are slightly more difficult to interpret as there remains a functional catalytic domain in the complex. This explains the fact that the EXT1 D565N/D567N and EXT2 D538N/D540N mutants have a less severe effect in the in cellulo experiments. The same observation is made for the EXT2 D538N/D540N mutant in our in vitro experiments. There are three possible explanations that no activity is observed in vitro for the EXT1 D565N/D567N mutant. First, the purified enzyme complex is less stable as the other complexes as found during nano differential scanning fluorimetry experiments (Supplementary Fig. 9). A decreased thermal stability could result that the protein partially aggregates during enzyme activity assays. Importantly, we see residual GlcNAc-T transferase activity in the EXT1 D565N/D567N mutant using longer incubation times. Another explanation could be, that the efficiency of EXT1 and EXT2 to catalyze GlcNAc transfer depends on the substrate. While a short dp8 oligo was used for the in vitro activity tests, the “native” substrate is used in the in cellulo experiments. Furthermore, other enzymes of the HS biosynthesis pathway, which also feature GlcNAc-T activity (e.g. EXTL3) could partially compensate the lower activity of EXT1-EXT2 complex.

An effect of N- and C-terminal tags cannot be excluded, but as we compare the activity of the mutants and WT in each system, harboring exactly the same tag combinations, it is unlikely that this plays a dominant role.

To clarify the interpretation of GlcNAc-T catalytic mutants, the following statement has been inserted into the manuscript: “A decrease in GlcNAc glycosyl transferase activity in the EXT1 D565N/D567N and EXT2 D538N/D540N mutants confirms that the C-terminal GT-A domains of EXT1 and EXT2 harbor GlcNAc-T activity and emphasize the important role of the Dx D motif (Fig. 4b).”

“While the in cellulo experiments showed a lower amount of HS for the EXT2 D538N/D540N mutant, we observed a stronger effect for the EXT1 D565N/D567N mutant in vitro. This difference could be explained by the slightly reduced thermal stability of the purified EXT1 D565N/D567N protein complex.”

Substrate binding to EXT2 GlcA-T, EDF 9: The text states that substrate binding to the domain is likely prevented by ‘an anti-parallel beta-sheet intruding into the donor substrate binding pocket’. Based on all figures showing this hairpin, it seems to be pointing away from the putative binding site. Hence, this argument should be revised or explained in more detail.

Our answer: We are thankful for this suggestion. We have added two panels to Supplementary Fig. 6 to illustrate how the anti-parallel beta-sheet occupies the UDP binding site in EXT2 and how this prevents donor substrate binding in this position.

Proposed reaction mechanism: All proposed reaction mechanisms (SN2 versus SN1) require knowledge of the acceptor binding sites as well as the base catalyst (at least for SN2). The manuscript fails to discuss any mode of acceptor binding, although the authors synthesized the dp8

and dp9 oligosaccharides that serve as acceptors in vitro. Insights into acceptor binding are required to validate any mechanistic insights.

Our answer: We thank the reviewer for pointing out the importance of additional analysis on acceptor binding. In order to obtain additional structural insight into how the acceptor substrates are bound in the EXT1-EXT2 complex, we have prepared a new cryo-EM sample, in which the EXT1-EXT2 complex was mixed with two acceptor oligosaccharides. Rather than using the fluorophore-containing dp8 and dp9 substrates used in the in vitro assays, we chose commercial dp7 and dp8 acceptor substrates (IDURON, UK) that contain a p-nitrophenyl group at the reducing end and that have a GlcA and GlcNAc at the non-reducing end, respectively. We collected a data set containing 10929 images on a 300 kV Titan Krios TEM (at ESRF, Grenoble) and obtained a 3D reconstruction at a nominal resolution of 3.2Å with a highly similar overall architecture compared to our UDP-bound structure. Despite adding the acceptor substrates for the GlcNAc-T and GlcA-T reaction, we did not observe any additional densities in any of the four catalytic sites (see also response to review #1). A reason for this could be that acceptor substrate binding occurs only transiently or that the binding affinity of the used acceptor substrates is very low. Despite our additional effort, we still do not have any structural insight into how the acceptor substrate is exactly bound. Nevertheless, we believe our structural and functional data provide sufficient insight to propose a catalytic mechanism for each of the activities catalyzed by the EXT1-EXT2 complex.

Therefore, we further examined our UDP-bound structure and used structural comparison to draw additional conclusions regarding the reaction mechanism.

An important feature of S_N2 -like inverting GTs is the presence of a base catalyst, which deprotonates the C4-hydroxyl group of the acceptor substrate to allow a nucleophilic attack on the C1 of the donor substrate. Structural comparison of EXT1 and POGLUT1 suggests, that either Asp162 or Asp164 might play this role in EXT1 (see new Supplementary Fig. 7, panel e). The importance of these aspartate residues is emphasized by our functional studies, which show that these residues are essential for EXT1 GlcA-T activity. The fact that these aspartate residues are already “mutated” to asparagines in the EXT2 pseudo-GlcA-T domain further supports our hypothesis that EXT2 has no GlcA-T activity.

Another important feature required for an S_N2 -like reaction mechanism is the presence of amino acid residues that stabilize the leaving group. The arginines Arg280 and Arg346 of EXT1 are in close proximity to the “pyrophosphate” leaving group and our mutational analysis suggests that Arg346, which is not conserved in EXT2, is important for catalytic activity. These observations are in favor with a S_N2 -like reaction mechanism, but a S_N1 -like reaction mechanism cannot be ruled out at this stage.

Given the high structural similarities between the GlcNAc-T active sites of EXT1 and EXT2 with EXTL2, we expect them to follow the S_{Ni} -like reaction mechanism of retaining glycosyltransferases.

To meet the reviewer’s concern and simultaneously explore our data in order to provide insight into the reaction mechanism, we added a panel to Supplementary Fig. 8e and reformulated the text in section “Substrate recognition in the four active sites”. We also added the following paragraph into the “Proposed mechanism for chain polymerization” section:

“The high structural similarities of the GlcNAc-T active sites of EXT1 and EXT2 with EXTL2, support a S_{Ni} -like reaction mechanism for GlcNAc transfer. Currently, there are no structural homologs of the GlcA-T domain of EXT1 and EXT2 known. Nevertheless, we propose that GlcA transfer is based on a S_N2 -like reaction mechanism. An important feature of S_N2 -like inverting GTs is the presence of a base catalyst, which deprotonates and thus activates the C4-hydroxyl group of the acceptor substrate for its nucleophilic attack on the C1 carbon atom of the donor substrate⁴⁰.”

Structural comparison of EXT1 GlcA-T and POGLUT1 suggests that either Asp162 or Asp164 might play this role (Supplementary Fig. 7). The arginine residues Arg280 and Arg346 are in close proximity to the pyrophosphate moiety of the UDP and are likely to stabilize the negative charge of the leaving group. Together, with the spatial constraints of the donor and acceptor binding pocket, our observations favor a S_N2-like mechanism. ”

Minor points:

The manuscript lacks discussions of previous data on EXT1 and EXT2 purification and function, showing that both enzymes can indeed be purified separately. Further, previous data shows that the EXT1/EXT2 heterodimer has much higher GlcA-T activity than it does GlcNAc-T activity (McCormick et al, 2000; Busse & Kusche-Gullberg, 2003). This is not discussed at all.

Busse & Kusche-Gullberg (2003) describe the preparation of protein extracts enriched in EXT1 and EXT2 proteins. The purity of the protein preparations is not shown in their manuscript and they have only small amount of enzymes in the nanogram range as judged by western blot analysis. Although we state in our manuscript that “expression of either EXT1 or EXT2 alone results in almost undetectable amounts of secreted protein”, this does not mean that it is impossible to obtain small amounts of the single proteins. However, we provide several lines of evidence that the great majority of our purified protein is present as hetero-dimeric complexes and we do not observe any homomeric complexes in our experiments.

We have added the following statement to the manuscript: “*Although previous studies suggested that EXT1 and EXT2 can be expressed on their own^{21,24}, we were only able to purify the complex upon co-expression of both proteins.*”

EXD 5c and d: Please state what glycans have been modeled.

Our answer: Modified as requested.

EXD 6b and d: Please define the distances of the interactions shown as dashed lines.

Our answer: Modified as requested.

Fig. 3C: The Y-axis requires a label (normalized abundance?).

Our answer: Modified as requested.

FACE electrophoresis: FACE was invented in the 90s. This should be acknowledged.

Our answer: Modified as requested.

Figure 4: As shown the figure is incorrect as it misses the tetrasaccharide linker region.

Our answer: We have modified Figure 4 to clarify the proposed mechanism and to emphasize the non-processivity of chain elongation.

REVIEWER COMMENTS

Reviewer #3 (Remarks to the Author):

The revised manuscript by Francisco Leisico et al. addressed some of my earlier comments, yet some significant concerns remain.

First regarding the discussed GlcAT reaction mechanism: As stated in my earlier review, discussing potential reaction mechanisms really requires knowledge of the donor and acceptor binding poses and coordination. Because this information is currently not available, the authors added another panel to SF7 showing potential base catalysts for the glycosyl transfer reaction. However, these Asp side chains seem too far away to be likely candidates, unless additional conformational changes are assumed that would bring the donor and acceptor closer together, and thereby also favoring proton transfer to the catalytic base. With the available information, any inferred reaction mechanism is highly speculative. Therefore, this discussion would be best provided as a supplementary discussion, but certainly not as a main conclusion based on the presented data.

Supplementary Fig. 6, UDP density: I am still concerned about the poor map quality representing the nucleotide. The slightly enlarged EM map in SF6a shows several poorly resolved side chains (F345, Y271, K269, K267), which is surprising for ligand-coordinating residues and an overall estimated resolution of 2.8 Å. I also find the argument that the furanose ring may oscillate between different conformations when bound to a catalytic pocket not particularly convincing. I am wondering whether the nucleotide is just not bound properly. Accordingly, the correct donor (and acceptor) positions would be unknown, further complicating any conclusions regarding the reaction mechanism.

EXT2 activity and mutagenesis:

It is stated in the abstract that 'EXT2 harbors only N-acetylglucosamine activity' and in the discussion that 'the EXT2 pseudo-GlcA-T domain is catalytic inactive and that glucuronic acid transfer is carried out solely by EXT1'. However, the strength of the biochemical evidence presented does not seem to support the strength of these statements. The apparent lack of in vitro GlcA-T activity for the D565N/D567N EXT1 mutant could perhaps be explained by reduced stability of the complex. Furthermore, since Arg280 is not fully essential for EXT1 GlcA-T activity, the fact that mutating its equivalent in EXT2 (Arg266) has little effect on GlcA-T activity is not so informative. Hence, these statements ought to be hedged. Alternatively, point mutants could be made for Glu349 in EXT1 and its equivalent Asp333 in EXT2 (with reference to GT-B donor binding motifs).

The authors should also mention/discuss recent work from Paul Dupree's group on EXTL3 and how their interpretations provide insights into EXT1/2 structure & function.

The time course in Figure 5b is a good addition to the manuscript, strengthening the distributive model for HS backbone extension. However, the oligosaccharide products are not labelled. Does the gap between bands correspond to one, or two monosaccharide units? Previous work has indicated that the GlcA-T activity of the EXT1/EXT2 complex is much stronger than its GlcNAc-T activity (e.g. McCormick et al 2000, Busse et al 2003...), which would suggest that the predominant products should be of even-numbered DPs (i.e. with GlcA at the non-reducing terminus), but this is not discussed. Finally, the references to Fig. 5b and Fig. 5a appear to have been reversed in the main text.

Alongside the cryo EM workflow, the manuscript lacks a figure showing representative parts of the map overlaid with the model. I apologize that I missed this earlier. This is important to allow non-experts to judge the map quality and thus confidence in the model.

Article “Structure of the human heparan sulfate polymerase complex EXT1-EXT2”

Answer to Reviewers

Reviewer #3 (Remarks to the Author):

The revised manuscript by Francisco Leisico et al. addressed some of my earlier comments, yet some significant concerns remain.

- First regarding the discussed GlcAT reaction mechanism: As stated in my earlier review, discussing potential reaction mechanisms really requires knowledge of the donor and acceptor binding poses and coordination. Because this information is currently not available, the authors added another panel to SF7 showing potential base catalysts for the glycosyl transfer reaction. However, these Asp side chains seem too far away to be likely candidates, unless additional conformational changes are assumed that would bring the donor and acceptor closer together, and thereby also favoring proton transfer to the catalytic base. With the available information, any inferred reaction mechanism is highly speculative. Therefore, this discussion would be best provided as a supplementary discussion, but certainly not as a main conclusion based on the presented data.

Our answer: We thank Reviewer #3 for pointing this out. We rephrased our description and interpretation of the EXT1 GlcA-T reaction mechanism in the section “proposed mechanism for chain polymerization” to make it clear, that this is hypothetical. And in the “conclusion” section we clearly state now that in light of the current data we can only presume that GlcA transfer follows a S_N2-like reaction mechanism. We did not add an additional “supplementary discussion” section to the supplementary material document, since this only contains illustrations and tables. However, we clearly tuned down our conclusions in the main text to address the concerns of Reviewer #3.

- Supplementary Fig. 6, UDP density: I am still concerned about the poor map quality representing the nucleotide. The slightly enlarged EM map in SF6a shows several poorly resolved side chains (F345, Y271, K269, K267), which is surprising for ligand-coordinating residues and an overall estimated resolution of 2.8 Å. I also find the argument that the furanose ring may oscillate between different conformations when bound to a catalytic pocket not particularly convincing. I am wondering whether the nucleotide is just not bound properly. Accordingly, the correct donor (and acceptor) positions would be unknown, further complicating any conclusions regarding the reaction mechanism.

Our answer: We agree with Reviewer #3. The map quality on the protein surface (including the catalytic sites) is much poorer than in the protein core and thus does not reflect the resolution of 2.8 Å, which is the average resolution for the full complex. We have clearly stated this observation in the main text and the method section of our manuscript. In Supplementary Fig. 7A, we show the EM density and the structural model surrounding the UDP ligand, including residues that are not involved in ligand coordination (F345, Y271, K267 and K269). Importantly, none of these residues are suggested to be involved in ligand binding in the manuscript. These residues seem to be surface exposed and are thus not expected to be well ordered and visible.

In contrast, the residues involved in ligand binding, such as R280, R346 and Y324 have well resolved side chains.

The observation that UDP binding sites are similar in EXT1 and POGLUT1 (Supplementary Fig. 8e,f) gives us further confidence, that the UDP ligand has been positioned correctly.

We introduced the concept of flexibility within the UDP furanose ring as one possibility to explain why the EM map is poorer in this region, without excluding other possibilities.

We hope that access to the experimental EM map will clear out doubts about positioning the UDP ligand.

- EXT2 activity and mutagenesis:

It is stated in the abstract that 'EXT2 harbors only N-acetylglucosamine activity' and in the discussion that 'the EXT2 pseudo-GlcA-T domain is catalytic inactive and that glucuronic acid transfer is carried out solely by EXT1'. However, the strength of the biochemical evidence presented does not seem to support the strength of these statements. The apparent lack of *in vitro* GlcA-T activity for the D565N/D567N EXT1 mutant could perhaps be explained by reduced stability of the complex. Furthermore, since Arg280 is not fully essential for EXT1 GlcA-T activity, the fact that mutating its equivalent in EXT2 (Arg266) has little effect on GlcA-T activity is not so informative. Hence, these statements ought to be hedged. Alternatively, point mutants could be made for Glu349 in EXT1 and its equivalent Asp333 in EXT2 (with reference to GT-B donor binding motifs).

Our answer: The EXT1 D565N/D567N mutant displays reduced GlcNAc-T, not GlcA-T, activity *in vitro*. We agree that this reduction in activity could be explained by a reduced stability of the complex and this was indicated in the manuscript (see lines 225-226).

Regarding the lack of GlcA-T activity in EXT2, our conclusion is supported by several lines of evidence, including i) strongly reduced *in vitro* activity observed for the EXT1 D162N/D164N and R280A mutants (with comparable thermal stability than WT complex in nanoDSF experiments); ii) almost abolished HS biosynthesis *in cellulose* for the EXT1 D162/D164 mutant; and iii) the absence of these residues in EXT2 and the presence of an anti-parallel beta-sheet blocking access to the active site. We report, in addition, that single point mutations in the EXT1 GlcA-T site are sufficient to confer HME skeletal disorder in human patients. Presumably, because EXT2 has no GlcA-T activity itself. This is in line with the fact that no deleterious mutations were described in the EXT2 GlcA-T active site. We thus believe that the ensemble of our data and observations strongly supports the inactivity of the EXT2 GlcA-T domain. However, we further tuned down our statements in the abstract and conclusion on EXT2 inactivity.

The authors should also mention/discuss recent work from Paul Dupree's group on EXTL3 and how their interpretations provide insights into EXT1/2 structure & function.

Our answer: We agree with Reviewer #3. We included a comparison between the homo-dimeric EXTL3 and EXT1-EXT2 structures in the manuscript and added a new supplementary figure (Supplementary Figure 10). We kept the substrate-bound EXTL2 crystal structure as the model reference for the description of GlcNAc-T active sites of EXT1 and EXT2 as it provides more mechanistic insights.

The time course in Figure 5b is a good addition to the manuscript, strengthening the distributive model for HS backbone extension. However, the oligosaccharide products are not labelled. Does the gap between bands correspond to one, or two monosaccharide units? Previous work has indicated that the GlcA-T activity of the EXT1/EXT2 complex is much stronger than its GlcNAc-T activity (e.g. McCormick et al 2000, Busse et al 2003...), which would suggest that the predominant products should be of even-numbered DPs (i.e. with GlcA at the non-reducing terminus), but this is not discussed. Finally, the references to Fig. 5b and Fig. 5a appear to have been reversed in the main text.

Our answer: We thank Reviewer #3 for this comment. It would be indeed very interesting to know, which different oligosaccharides have been generated, however, we currently don't have the suitable markers to address this question. But we are planning to investigate the kinetics of GlcA-T and GlcNAc-T reaction in future studies. In this manuscript, it seems too speculative to us to draw

conclusions about the catalytic rates of the different domains and the identity of the generated oligosaccharide lengths.

We corrected the references to Fig. 5b and Fig.5a – thanks for having noticed this mistake.

Alongside the cryo EM workflow, the manuscript lacks a figure showing representative parts of the map overlaid with the model. I apologize that I missed this earlier. This is important to allow non-experts to judge the map quality and thus confidence in the model.

Our answer: We prepared a new supplementary figure (Supplementary Fig. 4) illustrating varying map qualities in different regions of the map.

Additional feedback from Rev#3:

The provided Ext1-Ext2 map reveals significant uncertainties regarding the placement of side chains and backbone regions within the EXT1 GlcA-T domain. For example, lysine 267 and 269 are not resolved at all, of which K267 points into a density blob adjacent to Y319 that is essentially not accounted for. Y271, modeled to contact UDP's beta-phosphate, is extended by significant density that is unlikely to represent the side chain's hydroxyl group. The placement of the nucleotide's diphosphate group is speculative at best. The alpha-phosphate currently sits in a density sufficiently broad to accommodate both phosphates, while the beta-phosphate occupies a minor protrusion thereof, which could perhaps represent a water or associated ion. The density interpreted as the uracil moiety is much larger than the uracil itself, thus rendering its placement uncertain.

In summary, the map quality in its current form does not allow accurate modeling of the UDP ligand. This should be resolved, or the corresponding discussion be removed.

Our answer: To better demonstrate ligand localization and coordination, we have added an additional rotated view on the UDP ligand binding site to Supplementary Fig.7 (previously 6). Also, two more panels displaying the residues close to the active site were added to Supplementary Figure 4. Further, we have described the UDP binding more carefully in the manuscript and we emphasize more the weaknesses of the EM map in this region (lines 146-156).

I am wondering whether the map represents an average of different states. Perhaps a 3D classification and local refinement of this domain alone would resolve some of the ambiguities. Other regions of concern include Cys312 to K321 (the helix may be out of register) and L272 to D278, among various other side chain conformers.

Considering the variability in map quality, the authors should also explain in more detail all postprocessing steps applied to improve the map quality in the methods and Figure S3.

Our answer: We tried to improve the quality of the EM map in the UDP ligand binding site using a local refinement strategy. For this, we performed a 3D refinement job using a mask governing both GlcA-T domains. The obtained maps were of slightly poorer quality than when performing alignments for the full complex, probably due to the rather small size of this portion of the protein (Figure 1). We further performed 3D classifications and subsequent 3D refinements, however, obtained classes had only minor structural differences. As the number of particles was reduced, the

final resolution got worse. We extended the description of data processing in the method section (line 438-439).

Figure 1: Local refinement of GlcA-T domains of EXT1-EXT2 complex. Left: Resulting EM map with docked model. Right: Zoom-in into active site of EXT1 GlcA-T.